# FLEXIBLE PARTICIPATION FOR DIFFERENTIALLY PRIVATE SYNTHETIC TEXT GENERATION IN CROSS-SILO FEDERATED LEARNING

## ABSTRACT

In cross-silo federated learning (FL), sensitive text datasets remain confined to local organizations due to privacy regulations, making repeated training for each downstream task both communication-intensive and privacy-demanding. A promising alternative is to generate differentially private (DP) synthetic datasets that approximate the global distribution and can be reused across tasks. However, pretrained large language models (LLMs) often fail under domain shift, and federated finetuning is hindered by computational heterogeneity: only resource-rich clients can update the model, while weaker clients are excluded, amplifying data skew and the adverse effects of DP noise. We propose a flexible participation framework that adapts to client capacities. Strong clients perform DP federated finetuning, while weak clients contribute through a lightweight DP voting mechanism that refines synthetic text. To ensure the synthetic data mirrors the global dataset, we apply control codes (e.g., labels, topics, metadata) that represent each client's data proportions and constrain voting to semantically coherent subsets. This two-phase approach requires only a single round of communication for weak clients and integrates contributions from all participants. Experiments show that our framework improves distribution alignment and downstream robustness under DP and heterogeneity.

## 1 INTRODUCTION

In cross-silo federated learning (FL), sensitive text data are distributed across organizations and must remain local due to privacy regulations (Huang et al., 2022). Each client (e.g., a hospital, company, or organization) often stores thousands to tens of thousands of text samples collected from individuals (Sheller et al., 2018; Dayan et al., 2021), making it essential to train models collaboratively without sharing raw data. However, each downstream task typically requires initiating a new FL process, which incurs substantial communication overhead, additional privacy cost, and places extra burden on compute-constrained clients. A promising alternative is to generate synthetic datasets that act as privacy-preserving surrogates of the global dataset, thereby reducing both communication and privacy risks (Stadler et al., 2022; Yoon et al., 2020; Little et al., 2023). The objective of this work is to generate high-quality synthetic text that faithfully reflects the global distribution in cross-silo FL while providing rigorous differential privacy guarantees.

A straightforward solution is to directly generate texts from a pretrained language model (Hou et al., 2024) in FL. However, in many practical scenarios, such text exhibits low quality because the pretrained distribution diverges from the target global distribution. This issue arises, for example, when data distributions evolve over time or when domain adaptation is required (Gururangan et al., 2020; Cohen-Wang et al., 2024; Arakelyan et al., 2023). In this case, finetuning is essential to adapt the model for high-quality text generation.

Finetuning large language models (LLMs) in federated settings, however, faces a critical obstacle: computational heterogeneity (Bai et al., 2024; Liu et al., 2025; Wang et al., 2025). LLM finetuning demands substantial local resources, yet many clients in cross-silo FL lack the necessary computing capacity. As a result, only a fraction of clients with strong computing capacity can participate in model updates in a timely manner. This imbalance exacerbates the effects of data heterogeneity, as the global model is skewed toward the distributions of stronger clients while underrepresenting weaker ones. The situation is further worsened when differential privacy (DP) is enforced: DP-

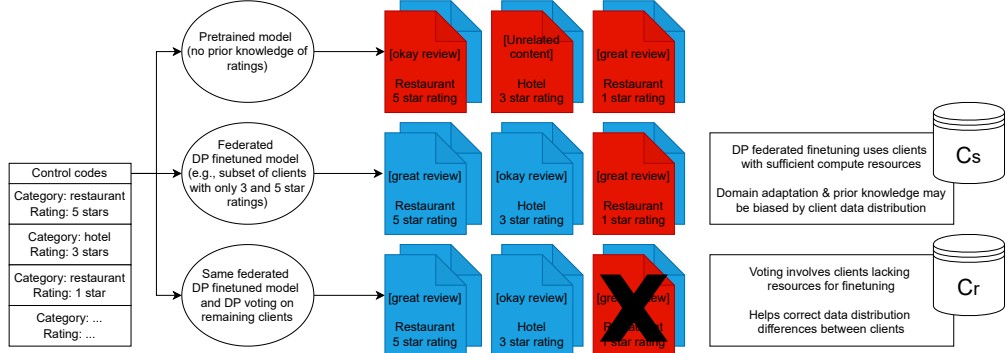

Figure 1: To perform DP synthetic text generation in cross-silo FL, we aim to address two challenges: heterogeneity in computational resources and data distributions. Our approach enables flexible participation through DP federated finetuning of the generator model on well-resourced clients and DP voting on generated synthetic text on the remaining clients.

SGD (Abadi et al., 2016) protects individual samples by injecting random noise into local updates. Reduced participation in local training can amplify the negative effect of DP noise, which hampers convergence and further degrades the quality of generated text (Wei et al., 2020).

To address these challenges brought by computational heterogeneity, we propose a flexible participation strategy that adapts to the computational capacities of clients in cross-silo FL. While clients with sufficient resources still engage in DP federated finetuning of the global generative model, weaker clients—those unable to perform expensive local updates—contribute through a lightweight voting mechanism. The key insight is that even partial finetuning allows the model to capture essential language patterns, while the voting stage refines the generated text according to the local data on weaker clients so that the negative effect of biased finetuning can be mitigated. An illustration of the framework can be found in Figure 1.

A challenge lies in characterizing data distributions so that the final synthetic dataset can effectively mirror the global population. To solve this, we adopt control codes (Keskar et al., 2019) (e.g., labels, topics, or metadata) to explicitly structure the data. Control codes partition texts into semantically meaningful subsets and serve two key roles in our framework. First, they represent each client's local distribution through control code proportions, which guide the allocation of synthetic samples across codes. Second, they constrain voting to samples within the same control code, ensuring that refinement is based on semantically coherent and relevant texts.

Overall, our framework proceeds in two phases. **Phase 1 (DP federated finetuning)**: strong clients update the global model using DP-SGD, adapting it to domain-specific data while preserving privacy. This finetuned model, though imperfect, captures broad patterns of the data. **Phase 2 (refinement via DP voting)**: weak clients contribute indirectly by providing DP-perturbed control code profiles and casting votes on synthetic text samples generated under each control code. The server aggregates these noisy votes to reweight and resample candidates, producing a final synthetic dataset that better aligns with the global population. Importantly, this refinement requires no backward propagation and only a single round of communication, making it efficient and inclusive even for clients with limited resources.

We evaluate our approach on benchmark datasets under both IID and non-IID settings with DP. The results show that even with a small proportion of strong clients (1–10%), partial finetuning improves the quality of synthetic data over zero-shot generation from pretrained models. More importantly, the refinement stage consistently boosts performance, mitigating the negative effects of biased finetuning and DP noise.

## 2 BACKGROUND

### 2.1 FEDERATED LEARNING

In FL, $N$ clients collaboratively train a model $\theta$ to minimize the averaged loss function:

$$\min_{\theta} f(\theta) := \frac{1}{N} \sum_{i=1}^{N} f_i(\theta), \tag{1}$$

where $f_i(\theta) = \ell(\theta; D_i)$ is the local loss function of client $i$, and $D_i$ is the local data distribution of client $i$. At the beginning of the $r$-th training round, each participating client receives the current global model parameters $\theta^r$ from a central server. Each client then performs $\tau$ steps of local model updates based on its own data. After completing local training, clients send their model updates back to the server, which aggregates these updates to form a refined global model for the next round.

When finetuning LLMs in federated settings, computational and data heterogeneity present significant challenges. Computational heterogeneity restricts the timely participation of resource-constrained clients due to the high computational demand of LLM training. At the same time, this partial finetuning may lead to bias for the finetuned model. In our algorithm, the text generated by the biased model will be refined further using non-training method to balance the negative effect of data heterogeneity.

## 2.2 Differential Privacy

In this work, DP will be applied both in finetuning and in refinement to guarantee the privacy of local data. Below is the formal definition of differential privacy.

**Definition 1 (Differential Privacy (Dwork et al., 2014))** *A randomized algorithm $M : \mathcal{D} \to \mathcal{S}$ is $(\varepsilon, \delta)$ differentially private if any two neighboring datasets $D, D' \in \mathcal{D}$ that differ exactly in a single data sample, and for all sets $S \in \mathcal{S}$:*

$$\mathbb{P}[M(D) \in S] \leq e^\varepsilon \mathbb{P}[M(D') \in S] + \delta. \tag{2}$$

In the definition, $\varepsilon$ determines the privacy budget and $\delta$ is the probability of failure.

To protect the privacy of each individual, we consider sample-level DP, ensuring that the output of each client is independently perturbed for every data sample. In finetuning, DP-SGD (Abadi et al., 2016) is applied in local training on strong clients, where Gaussian noise is added to the gradients. In refinement, analytical Gaussian mechanism (Balle & Wang, 2018) is applied, where the profiles and votes of local data are perturbed.

## 3 Method

### 3.1 Problem Formulation

We consider text generation in the FL scenario, where each of the $N$ clients maintains a local text dataset $D_i, \forall i \in [N]$. Due to the data heterogeneity, the local datasets $D_i$ may differ substantially across clients. The global dataset is defined as $D := \cup_{i=1}^{N} D_i$. The objective is to generate a synthetic dataset $\tilde{D}$ that closely approximates the global dataset $D$ with DP guarantee. The setups of the clients and the server are detailed as follows:

**Clients.** Due to computational heterogeneity, only a subset of clients can efficiently perform local finetuning of LLMs in a timely manner. Let $\mathcal{C}_s \subseteq [N]$ denote the set of clients with sufficient computational resources, where $|\mathcal{C}_s| = M, 1 \leq M \leq N$. The set of remaining clients is denoted as $\mathcal{C}_r = [N] \setminus \mathcal{C}_s$. Although clients in $\mathcal{C}_r$ do not directly participate in federated finetuning, they can contribute indirectly by sending DP statistical profiles (e.g., summary statistics or votes) of their local data to the server for the potential improvement on the text generation.

**Server.** The server initializes and maintains a pretrained language model. Since the distribution of pretrained model $p_\theta(\cdot)$ might not be aligned with the current global dataset $D$ due to the domain difference or the distribution change over time, the pretrained model has to be finetuned over the current data for generating consistent text. The server coordinates the federated finetuning by aggregating private model updates from clients in $\mathcal{C}_s$ and broadcasting the latest global model to $\mathcal{C}_s$ clients. The server then leverages the resulting global model to produce synthetic text data representative of the global data distribution.

The main challenge is ensuring that the synthetic dataset captures the distributions of both $\mathcal{C}_s$ and $\mathcal{C}_r$ clients despite differences in data and computation. Federated finetuning enables the model to learn from $\mathcal{C}_s$ data, but the generated text may not be representative of the data from $\mathcal{C}_r$. The key question is how to also model the data from $\mathcal{C}_r$ without direct finetuning. In Section 3.2, we address this by using controllable text generation, allowing clients in $\mathcal{C}_r$ to guide the generation process via their statistical profiles.

**Algorithm 1:** Federated Controllable Text Generation with Refinement

---

**Input** : Control codes $C$ ; client sets $\mathcal{C}_s, \mathcal{C}_r$; initial model $\theta^0$; FL rounds $R$, local iteration $\tau$;
DP params: $(\varepsilon_{\text{train}}, \delta_{\text{train}}), (\varepsilon_{\text{prof}}, \delta_{\text{prof}}), (\varepsilon_{\text{vote}}, \delta_{\text{vote}})$; number of synthetic samples $s$,
sentence transformer $g(\cdot)$, sampling rate $r$, number of votes $K$.

**Output:** Synthetic dataset $\tilde{D}$.

**Stage 1: Federated DP finetuning on $\mathcal{C}_s$**

**Server:** $\theta^\star \leftarrow \texttt{Federated\_Finetuning}(\theta^0, \mathcal{C}_s, R, \tau, \varepsilon_{train}, \delta_{train})$ .

**Stage 2: DP profiling from clients**

**foreach** $i \in [N]$ **do**

    Compute control-code counts $P_i \leftarrow \left[|D_i^1|, \dots, |D_i^{|C|}|\right]$ with $D_i = \cup_{j=1}^{|C|} D_i^j$

    $\tilde{P}_i \leftarrow \texttt{Analytical\_Gaussian\_Mechanism}(P_i, \varepsilon_{prof}, \delta_{prof})$

**Server:** Form a global target profile $\tilde{P} \leftarrow \sum_{i=1}^N \tilde{P}_i$ .

**Stage 3: Synthetic generation guided by profiles**

**for** $j = 1$ **to** $|C|$ **do**

    For $\tilde{D}^j$, generate $s_j$ samples using model $p_{\theta^\star}(\cdot \mid c^j)$ with $s_j = \text{Round}(s \cdot \tilde{P}[j])$

**Server:** Set initial synthetic dataset $\tilde{D} \leftarrow \cup_{j=1}^{|C|} \tilde{D}^j$.

**Stage 4: DP voting-based refinement using $\mathcal{C}_r$**

**foreach** $i \in \mathcal{C}_r$ **do**

    $\tilde{v}_i \leftarrow \texttt{Local\_Voting}(\tilde{D}, D_i, C, g, K, \epsilon_{vote}, \delta_{vote})$

**Server:** Aggregate the votes $\tilde{v} \leftarrow \sum_{i=1}^N \tilde{v}_i$.

**for** $j = 1$ **to** $|C|$ **do**

    $p^j \leftarrow \tilde{v}[\mathcal{I}^j]/\|\tilde{v}[\mathcal{I}^j]\|_1$ with $\mathcal{I}^j = \{i | \tilde{D}[i] \in \tilde{D}^j\}$

    $\tilde{\mathcal{I}}_j \leftarrow \texttt{Sampling\_Without\_Replacement}(\mathcal{I}^j, r, p^j)$

**Return** $\tilde{D}^* \leftarrow \cup_{j=1}^{|C|} \tilde{D}[\tilde{\mathcal{I}}_j]$

---

### 3.2 Algorithm

In our algorithm, we consider conditional language modeling (Keskar et al., 2019):

$$p_\theta(x|c) = \prod_{l=1}^L p_\theta(x_l|x_{<l}, c), \tag{3}$$

where $L$ is the length of the sequence and $c$ is the control code. When generating texts after finetuning, the control code is used as the prompt for controllable generation. This approach offers two key benefits: (i) prompts do not need to be hand-designed, since the control code captures the training data distribution, and (ii) in FL, control codes can also represent local data distributions of clients in $\mathcal{C}_r$, enabling the generation of text corresponding to their profiles without further finetuning. The details of controllable generation in FL are as follows.

On client $i \in [N]$, the local data distribution can be decomposed by a set of control codes $C = \{c^1, c^2, \dots, c^{|C|}\}$. The set of control codes is situational dependent and may correspond to the labels, the topics, or the features of the training data. Following Yue et al. (2022), we assume that the control codes are not private. As mentioned above, all clients share the same set of the control codes $C$, which is predetermined. Each example in the local datasets is associated with exactly one control code $c \in C$ for local finetuning. We use $D_i^j$ to denote the set of local data related to the control code $c^j$ on client $i$ and we have $\cup_{j=1}^{|C|} D_i^j = D_i$. Then given $C$, we can represent the distribution of the local dataset using the following vector

$$P_i = \left[|D_i^1|, |D_i^2|, \dots, |D_i^{|C|}|\right], \forall i \in [N]. \tag{4}$$

The global distribution is $P = \sum_{i=1}^N P_i$. With control codes, data heterogeneity in FL can be expressed in a hierarchical manner. At the first level, the control code distributions $P_i$ vary across

clients, reflecting differences in how data categories are represented locally. These $P_i$ vectors are used to determine the amount of synthetic data to generate for each control code, ensuring that the synthetic dataset matches the overall distribution. At the second level, even within a given control code $c^j$, the underlying data across clients may differ. Since data from $\mathcal{C}_r$ clients are not directly involved in the federated finetuning process, we introduce a refinement step to better capture their distributions and reduce potential bias in the generated text after federated finetuning. The overall workflow is summarized in Algorithm 1, and its key stages are described in detail below.

**Federated Finetuning.** The local objective function of client $i \in \mathcal{C}_s$ is given by

$$f_i(\theta) = \mathcal{L}_\theta(D_i) = -\frac{1}{|D_i|} \sum_{j=1}^{|C|} \sum_{x \in D_i^j} \log p_\theta(x|c^j), \forall i \in \mathcal{C}_s, \tag{5}$$

where $\mathcal{L}_\theta(\cdot)$ is the loss function. During this stage, clients in $\mathcal{C}_s$ periodically perform local DP-SGD and send the local model to the server for the aggregation. This stage can be found in Algorithm A.3.

**Profiling.** After federated finetuning, the global model has captured knowledge of the current text pattern. To account for client-specific heterogeneity, the server generates synthetic text conditioned on local data profiles defined by control codes. Specifically, each client $i$ sends its profile vector $P_i$ perturbed by DP noise to the server, and the amount of text generated under each control code is proportional to the corresponding entries of $P_i$. The server then generates the initial synthetic texts according to $P_i$ and broadcasts them to $\mathcal{C}_r$ clients for refinement.

**Refinement.** The finetuned model may still struggle to generate high-quality synthetic data due to limitations inherent in FL. First, data heterogeneity implies that the local datasets of clients in $\mathcal{C}_s$ may be biased, preventing the model from fully representing the global distribution. Second, the application of DP-SGD during local updates introduces random noise, which can hinder convergence and further degrade text quality.

To mitigate these issues, we leverage the local data on clients in $\mathcal{C}_r$. Although they do not participate directly in federated finetuning, their local data can still influence text generation through a voting mechanism. The key idea is that, within each control code, each example of the local data on $\mathcal{C}_r$ casts $K$ votes for candidate synthetic samples generated under the same control code. After collecting all the votes, the analytical Gaussian mechanism is applied to guarantee DP. The aggregated DP votes is then used to resample and refine the synthetic data, aligning it more closely with the global data distribution. The pseudocode of local voting can be found in Algorithm A.2.

## 4 Experiments

**Datasets.** We evaluate our approach on two text corpora: Yelp Reviews (Yelp, Inc.) and PubMed abstracts (National Center for Biotechnology Information (NCBI)). Both datasets are partitioned into clients to simulate the cross-silo FL setting. Following the setup in Yue et al. (2022), we perform DP finetuning and synthetic text generation on the Yelp dataset, while PubMed serves as a domain-specific corpus to evaluate domain adaptation after finetuning. For Yelp, we use business categories and rating stars as control codes. For PubMed, we select five medical subject headings (MeSH terms from (Rogers, 1963)) as control codes and represent each abstract with a binary indicator (0/1) for whether it belongs to a given MeSH term. The five selected MeSH terms are Anatomy (abbreviated 'A'), Diseases ('C'), Chemicals and Drugs ('D'), Persons ('M'), and Healthcare ('N'). We keep a part of held-out data as the test datasets and evaluation datasets.

**Data Partition.** We consider both IID and non-IID partitionings. To be consistent with the cross-silo setting in FL, we partition each dataset into tens or one hundred clients, each with more that one thousand individual examples. Specifically, the Yelp dataset is partitioned into 100 clients, each with 15000 examples. The PubMed dataset is partitioned into 20 clients, each with 2250 examples. The percentage of $\mathcal{C}_s$ clients is varied across experiments. For IID cases, we uniformly partition both the Yelp and PubMed datasets into clients. For non-IID cases, we use different strategies for Yelp and PubMed according to their attributes. The goal is to show the data heterogeneity between $\mathcal{C}_s$ data and $\mathcal{C}_r$ data. In particular, for Yelp, when partitioning data for $\mathcal{C}_s$ clients, we fix one label then vary the number of classes of another label. For PubMed, we vary the number of MeSH terms covered by $\mathcal{C}_s$ data. Then we uniformly partition the remaining data into $\mathcal{C}_r$ clients.

Table 1: Experimental results for downstream tasks using Yelp synthetic data with IID setting. The results are partitioned into three parts according the conditions of DP and refinement. For each part, differenet percentages of $\mathcal{C}_s$ client are considered. Pre. means the synthetic data are directly generated from a pretrained model. Acc.-1 and F1-1 represent the accuracy and F1 score for category classification. Acc.-2 and F1-2 represent the accuracy and F1 score for rating classification.

| $\mathcal{C}_s\%$ | $\varepsilon = \infty$ | | | | $\varepsilon = 8 \downarrow$ | | | | $\varepsilon = 8$ with refinement $\uparrow$ | | | |
|---|---|---|---|---|---|---|---|---|---|---|---|---|
| | Acc.-1 | F1-1 | Acc.-2 | F1-2 | Acc.-1 | F1-1 | Acc.-2 | F1-2 | Acc.-1 | F1-1 | Acc.-2 | F1-2 |
| Pre. | 0.7044 | 0.6240 | 0.4414 | 0.2704 | — | — | — | — | 0.7356 | 0.6818 | 0.4632 | 0.3295 |
| 1% | 0.7367 | 0.7129 | 0.6541 | 0.6289 | 0.6815 | 0.6729 | 0.5113 | 0.3755 | 0.7126 | 0.6981 | 0.6149 | 0.5755 |
| 5% | 0.7485 | 0.7299 | 0.6644 | 0.6455 | 0.6968 | 0.6842 | 0.6145 | 0.5656 | 0.7110 | 0.6978 | 0.6277 | 0.5942 |
| 10% | 0.7487 | 0.7288 | 0.6661 | 0.6460 | 0.7123 | 0.6959 | 0.6280 | 0.5819 | 0.7252 | 0.7068 | 0.6326 | 0.6002 |
| 20% | 0.7469 | 0.7280 | 0.6707 | 0.6519 | 0.7158 | 0.6941 | 0.6328 | 0.5937 | 0.7247 | 0.7060 | 0.6464 | 0.6285 |
| 30% | 0.7458 | 0.7266 | 0.6717 | 0.6611 | 0.7130 | 0.6935 | 0.6306 | 0.6058 | 0.7243 | 0.7097 | 0.6470 | 0.6357 |
| 40% | 0.7474 | 0.7242 | 0.6744 | 0.6630 | 0.7261 | 0.7011 | 0.6428 | 0.6168 | 0.7349 | 0.7162 | 0.6446 | 0.6314 |

Table 2: Experimental results for downstream tasks using PubMed synthetic data with IID setting. Classification results for Chemicals and Drugs (D) and Healthcare (N) are provided. More classification results can be found in Section E.

| $\mathcal{C}_s\%$ | $\varepsilon = \infty$ | | | | $\varepsilon = 8 \downarrow$ | | | | $\varepsilon = 8$ with refinement $\uparrow$ | | | |
|---|---|---|---|---|---|---|---|---|---|---|---|---|
| | Acc.(D) | F1(D) | Acc.(N) | F1(N) | Acc.(D) | F1(D) | Acc.(N) | F1(N) | Acc.(D) | F1(D) | Acc.(N) | F1(N) |
| Pre. | 0.6456 | 0.5881 | 0.5364 | 0.5179 | — | — | — | — | 0.6464 | 0.5472 | 0.5864 | 0.5696 |
| 5% | 0.6672 | 0.6113 | 0.6996 | 0.6996 | 0.6220 | 0.5019 | 0.5908 | 0.5854 | 0.8028 | 0.8024 | 0.7368 | 0.7326 |
| 10% | 0.6904 | 0.6460 | 0.7164 | 0.7159 | 0.6332 | 0.5203 | 0.6076 | 0.6086 | 0.8100 | 0.8074 | 0.7348 | 0.7341 |
| 20% | 0.7968 | 0.7913 | 0.7140 | 0.7144 | 0.6304 | 0.5346 | 0.6220 | 0.6179 | 0.8268 | 0.8280 | 0.7368 | 0.7374 |
| 30% | 0.8612 | 0.8605 | 0.7432 | 0.7438 | 0.6380 | 0.5577 | 0.6284 | 0.6187 | 0.8312 | 0.8318 | 0.7400 | 0.7406 |
| 40% | 0.8788 | 0.8788 | 0.7496 | 0.7498 | 0.6400 | 0.5765 | 0.6384 | 0.6293 | 0.8426 | 0.8448 | 0.7436 | 0.7442 |

**Models.** We finetune GPT-2 (Radford et al., 2019) to generate synthetic Yelp reviews. We finetune GPT-2-large to generate synthetic PubMed abstracts for its better capacity of domain adaptation. We use stsb-roberta-base-v2 (Reimers & Gurevych, 2019) to generate the sentence embedding used in refinement for both Yelp and PubMed. We use RoBERTa-base (Liu et al., 2019) to perform downstream tasks for Yelp dataset. We use BERT (Devlin et al., 2019) to perform downstream tasks for PubMed.

**Metrics.** We evaluate the quality of synthetic data from two perspectives: (i) downstream utility, measured by classification accuracy and F1 score on tasks trained with synthetic data, and (ii) distributional alignment, measured by similarity between original and synthetic text as well as domain adaptation performance.

For Yelp, we evaluate downstream utility by reporting classification accuracy and F1 score on ratings and categories with synthetic reviews using RoBERTa. We evaluate the distributional alignment by computing the MAUVE score (Pillutla et al., 2021) between original and synthetic texts using GPT-2/GPT-2-large embeddings. For PubMed, we evaluate downstream utility by reporting classification accuracy and F1 score on medical categories with synthetic abstracts using BERT. We evaluate domain adaptation by reporting the macro F1 score for medical named entity recognition (NER) task using BERT.

**Baselines.** We compare against two main baselines. (i) To quantify the effect of partial finetuning, we report results from pretrained models without any finetuning. Since prompt design and control codes are orthogonal, and can even be combined as shown by Keskar et al. (2019), we adopt zero-shot generation with pretrained models for a fair comparison. (ii) To isolate the impact of differential privacy, we additionally provide results from models trained without DP. Together, these baselines allow us to evaluate both the benefits of finetuning and the trade-offs introduced by DP. Further details on the experimental setup and hyperparameters are provided in the Appendix.

## 4.1 IID RESULTS

We first analyze the performance of our algorithm under the IID setting. In particular, we investigate the effect of varying percentages of $\mathcal{C}_s$ clients and the refinement step. We show that even partial finetuning when only 1% of $\mathcal{C}_s$ clients participate can improve the data quality over the pretrained model and the refinement step can mitigate the negative effect of DP significantly.

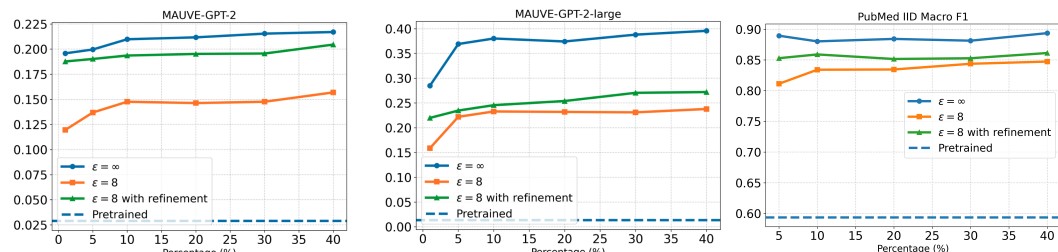

Figure 2: MAUVE score for Yelp IID results and NER macro F1 score for PubMed IID results.

Table 1 reports the two downstream task results using Yelp synthetic data: business category classification and rating classification. Overall, as we add more training data, we observe that increasing the $C_s$ clients percentage consistently improves both accuracy and F1 score. Notably, performance saturates quickly: even with only 10% $C_s$ clients, the accuracy and F1 scores become comparable to higher $C_s$ clients levels.

Comparing the results across privacy settings, DP leads to a substantial performance drop relative to the $\varepsilon = \infty$ baseline, due to the random noise introduced by DP-SGD. However, our refinement step mitigates this effect, yielding consistent improvements. For instance, at just 1% $C_s$ clients, refinement improves rating classification accuracy by 0.1 and F1 score by 0.2, making the performance comparable to 10% $C_s$ clients under DP without refinement. Similarly, with 20% $C_s$ clients and refinement, accuracy and F1 exceed those of 40% $C_s$ clients under DP without refinement. These findings demonstrate that even a single round of refinement substantially reduces the negative impact of DP noise. A closer look at the refinement behavior via voting statistics is shown in Figure A.3.

Table 1 also highlights differences in prior knowledge embedded in the pretrained model between the business category and rating tasks. Without DP, the pretrained model already achieves competitive accuracy on business category classification, surpassing the 1% and 5% $C_s$ clients cases under DP without refinement. However, for the rating prediction task (Acc.-2 and F1-2), the pretrained model essentially defaults to predicting the majority class, which accounts for 44% of the data, leading to poor F1 performance. After refinement, the pretrained model's accuracy on the business categories task even surpasses that of the 30% $C_s$ clients case with refinement, while its performance on ratings remains below all other settings. This suggests that the pretrained model carries useful prior knowledge for the majority classes in the business categories task, but not for minority business category classes or the ratings task, and that low-$C_s$-percentage DP finetuning can sometimes erode this prior knowledge.

Table 2 shows experimental results with synthetic data generated from PubMed for two downstream tasks: Chemicals and Drugs (D) and Healthcare (N) medical subject classification. In the $\varepsilon = \infty$ baseline setting, accuracy improves with the increased percentage of $C_s$ clients for both tasks but does not saturate as quickly as with Yelp. Adding DP with $\varepsilon = 8$ dramatically decreases learning, with accuracy at 40% $C_s$ clients in federated finetuning not matching 5% $C_s$ clients on the baseline. However, applying refinement to the $\varepsilon = 8$ case changes this behavior; not only does accuracy significant improve across both tasks but it exceeds the $\varepsilon = \infty$ baseline for low rates of $C_s$ clients. For instance, performance at 5% $C_s$ clients for $\varepsilon = 8$ with refinement exceeds that of 20% $C_s$ clients for $\varepsilon = \infty$. Although this trend does not persist at higher $C_s$ percentages, $\varepsilon = 8$ with refinement remains competitive with the baseline for the Chemicals and Drugs classification task and matches it for Healthcare classification. Results from the other three downstream tasks are shown in Section E and exhibit broadly similar trends.

Figure 2 reports MAUVE scores for Yelp and NER macro F1 score for PubMed. Results without DP serve as an upper bound, while with DP the scores consistently improve after refinement. For Yelp, both GPT-2 and GPT-2-large show that fidelity increases with finetuning and more $C_s$ clients. Notably, the evaluation method affects interpretation: with GPT-2, DP with refinement appears comparable to non-private generation, but GPT-2-large reveals persistent gaps, which might be due to the capacity of the larger model can capture more nuance. For PubMed, only the scores for DP results are consistently increasing with additional $C_s$ clients, indicating that under DP, only 5% $C_s$ clients with refinement can achieve competitive scores in this case.

Table 3: Experimental results for downstream tasks using Yelp synthetic data with non-IID setting. Two classes of experiments are presented; one where the rating classes between clients are varied and one where the business category classes between clients are varied.

| Rating classes | Category classes | $\varepsilon = \infty$ | | $\varepsilon = \infty$ with refinement | | $\varepsilon = 8$ | | $\varepsilon = 8$ with refinement | |
|---|---|---|---|---|---|---|---|---|---|
| | | Acc.-2 | F1-2 | Acc.-2 | F1-2 | Acc.-2 | F1-2 | Acc.-2 | F1-2 |
| 1 & 3 stars | All | 0.5394 | 0.4007 | 0.5898 | 0.4899 | 0.5266 | 0.3895 | 0.5790 | 0.4778 |
| 1 & 5 stars | All | 0.5701 | 0.4971 | 0.6234 | 0.6017 | 0.5658 | 0.4853 | 0.6008 | 0.5876 |
| 3 & 5 stars | All | 0.5748 | 0.5394 | 0.6023 | 0.6116 | 0.5632 | 0.5155 | 0.5904 | 0.5828 |
| 1, 3, & 5 stars | All | 0.6475 | 0.6174 | 0.6563 | 0.6321 | 0.6084 | 0.5740 | 0.6232 | 0.6078 |
| Rating classes | Category classes | $\varepsilon = \infty$ | | $\varepsilon = \infty$ with refinement | | $\varepsilon = 8$ | | $\varepsilon = 8$ with refinement | |
| | | Acc.-1 | F1-1 | Acc.-1 | F1-1 | Acc.-1 | F1-1 | Acc.-1 | F1-1 |
| All | 2 Categories | 0.7006 | 0.6699 | 0.7064 | 0.6768 | 0.6713 | 0.6540 | 0.6762 | 0.6689 |
| All | 4 Categories | 0.7302 | 0.7021 | 0.7324 | 0.7083 | 0.7118 | 0.6896 | 0.7154 | 0.6932 |
| All | 6 Categories | 0.7315 | 0.7073 | 0.7335 | 0.7113 | 0.7123 | 0.6904 | 0.7153 | 0.6982 |
| All | 8 Categories | 0.7445 | 0.7178 | 0.7447 | 0.7241 | 0.7157 | 0.6936 | 0.7248 | 0.7040 |

Table 4: Experimental results for downstream tasks using PubMed synthetic data in non-IID setting. Classification results for Anatomy (A), Chemicals and Drugs (D) and Diseases (C) are provided in three settings where only the subset of classes list in the Bias column are represented on clients participating in federated finetuning. More classification results can be found in Section E.

| Bias | $\varepsilon = \infty$ | | $\varepsilon = \infty$ with refinement | | $\varepsilon = 8$ | | $\varepsilon = 8$ with refinement | |
|---|---|---|---|---|---|---|---|---|
| | Acc.(A) | F1(A) | Acc.(A) | F1(A) | Acc.(A) | F1(A) | Acc.(A) | F1(A) |
| M, N | 0.5588 | 0.5526 | 0.6324 | 0.6222 | 0.4827 | 0.4719 | 0.5246 | 0.5123 |
| C, M, N | 0.5812 | 0.5757 | 0.6536 | 0.6536 | 0.5128 | 0.5035 | 0.5835 | 0.5732 |
| D, C, M, N | 0.6420 | 0.6405 | 0.6724 | 0.6725 | 0.5729 | 0.5617 | 0.6139 | 0.6044 |
| Bias | $\varepsilon = \infty$ | | $\varepsilon = \infty$ with refinement | | $\varepsilon = 8$ | | $\varepsilon = 8$ with refinement | |
| | Acc.(D) | F1(D) | Acc.(D) | F1(D) | Acc.(D) | F1(D) | Acc.(D) | F1(D) |
| M, N | 0.5696 | 0.5217 | 0.6840 | 0.6819 | 0.5034 | 0.4688 | 0.5662 | 0.5337 |
| C, M, N | 0.6052 | 0.5770 | 0.7304 | 0.7254 | 0.5437 | 0.5264 | 0.6638 | 0.6429 |
| D, C, M, N | 0.6980 | 0.6984 | 0.7548 | 0.7544 | 0.6338 | 0.6135 | 0.7047 | 0.6828 |
| Bias | $\varepsilon = \infty$ | | $\varepsilon = \infty$ with refinement | | $\varepsilon = 8$ | | $\varepsilon = 8$ with refinement | |
| | Acc.(C) | F1(C) | Acc.(C) | F1(C) | Acc.(C) | F1(C) | Acc.(C) | F1(C) |
| M, N | 0.6628 | 0.6357 | 0.6720 | 0.6750 | 0.6023 | 0.5836 | 0.6435 | 0.6248 |
| C, M, N | 0.8656 | 0.8663 | 0.8732 | 0.8739 | 0.7824 | 0.7719 | 0.8237 | 0.8142 |
| D, C, M, N | 0.7580 | 0.7345 | 0.8400 | 0.8380 | 0.7021 | 0.6934 | 0.7835 | 0.7754 |

## 4.2 NON-IID RESULTS

In this section, we analyze the performance of our algorithm with non-IID setting. In particular, we fix the percentage of $\mathcal{C}_s$ clients as $10\%$ then investigate the performance of our algorithm under data heterogeneity and DP. Experimental results show that our algorithm can mitigate both the negative effect of data heterogeneity and of DP. Notably, in some cases, after refinement, the accuracy and F1 score of $\varepsilon = 8$ can be better than those of $\varepsilon = \infty$.

Non-IID results for downstream tasks with Yelp can be found in Table 3. As expected, privacy and data heterogeneity affect the utility of the synthetic data for downstream prediction. Overall, as $\mathcal{C}_s$ data become more diverse, the accuracy and the F1 score increase. It can be observed from $\varepsilon = \infty$ with refinement results that refinement can improve accuracy and F1 score deteriorated by data heterogeneity. In cases of rating class partition, except 1, 3, & 5 stars, the accuracy and the F1 score under DP with refinement can even be better than those without DP.

Similar trends are apparent in Table 4 for downstream tasks with PubMed. For classification for Anatomy (A), Chemicals and Drugs (D), and Dieases (C), increasingly diverse $\mathcal{C}_s$ data improves accuracy and F1 scores in the $\varepsilon = \infty$ baseline even when not related to the particular classification task. Refinement provides consistent improvements in the baseline case, particularly for Diseases classification. Scores for $\varepsilon = 8$ with refinement match or exceed the baseline for Disease, and perform nearly as well for the other two tasks.

Table 5 reports MAUVE scores for Yelp synthetic data under non-IID settings, which align with the corresponding classification results. Compared to the $\varepsilon = \infty$ baseline, applying DP moderately reduces scores across both models. Refinement without DP brings performance closer to IID levels (Figure 2), while refinement with DP yields substantial gains—matching the $\varepsilon = \infty$ baseline for non-IID rating classes but still trailing for non-IID category classes. Table 6 provided NER macro

Table 5: MAUVE score for Yelp non-IID setting evaluated with embeddings generated from pre-trained GPT-2 and GPT-2-large. The experimental setting is the same as the setting in Table 3.

| Rating classes | Category classes | $\varepsilon = \infty$ | | $\varepsilon = \infty$ with refinement | | $\varepsilon = 8$ | | $\varepsilon = 8$ with refinement | |
|---|---|---|---|---|---|---|---|---|---|
| | | GPT-2 | GPT-2-l | GPT-2 | GPT-2-l | GPT-2 | GPT-2-l | GPT-2 | GPT-2-l |
| **1 & 3 stars** | All | 0.1404 | 0.1430 | 0.1907 | 0.1969 | 0.1105 | 0.1180 | 0.1493 | 0.1552 |
| **1 & 5 stars** | All | 0.1617 | 0.2204 | 0.2128 | 0.2675 | 0.1444 | 0.1988 | 0.1915 | 0.2374 |
| **3 & 5 stars** | All | 0.1634 | 0.2312 | 0.2258 | 0.2785 | 0.1501 | 0.1921 | 0.1987 | 0.2372 |
| **1, 3, & 5 stars** | All | 0.1924 | 0.3465 | 0.2463 | 0.3658 | 0.1975 | 0.2425 | 0.1982 | 0.2600 |
| All | **2 Categories** | 0.1808 | 0.2912 | 0.2414 | 0.2808 | 0.1414 | 0.2016 | 0.1911 | 0.2174 |
| All | **4 Categories** | 0.2092 | 0.3520 | 0.2510 | 0.3640 | 0.1456 | 0.2122 | 0.1907 | 0.2410 |
| All | **6 Categories** | 0.2047 | 0.3381 | 0.2543 | 0.3702 | 0.1426 | 0.2110 | 0.1946 | 0.2474 |
| All | **8 Categories** | 0.2266 | 0.3487 | 0.2674 | 0.3751 | 0.1457 | 0.2298 | 0.1952 | 0.2586 |

Table 6: NER macro F1 score for PubMed non-IID results.

| Setting | $\varepsilon = \infty$ | | | $\varepsilon = 8$ | | |
|---|---|---|---|---|---|---|
| | M,N | C,M,N | D,C,M,N | M,N | C,M,N | D,C,M,N |
| **No refine / Refine** | 0.5783 / 0.6529 | 0.8344 / 0.8402 | 0.8641 / 0.8739 | 0.8035 / 0.8521 | 0.8052 / 0.8513 | 0.8199 / 0.8523 |

F1 scores for PubMed non-IID results. Refinement consistently improves the performance under DP and heterogeneity. However, it is worth noting that when only M, N data are on $\mathcal{C}_s$ clients, the scores of $\varepsilon = \infty$ are worse than those of $\varepsilon = 8$. One possible reason could be that the non-private model overfits their skewed distribution, whereas DP-SGD's clipping and noise act as implicit regularization for NER task.

## 5 RELATED WORK

**Federated learning.** Since the introduction of FL (McMahan et al., 2017), extensive work has addressed its two core challenges: data heterogeneity (Li et al., 2020; Karimireddy et al., 2020) and computational heterogeneity (Lai et al., 2021; Nguyen et al., 2022). Recently, for adapting LLMs in FL, parameter-efficient fine-tuning (PEFT) with adapters/LoRA (Wu et al., 2025) reduces compute and communication but still requires backpropagation on participating clients including FLoRA (Wang et al., 2024) and other approaches (Ghiasvand & colleagues, 2024; Hao et al., 2024). However, these approaches typically presume broad client participation for finetuning.

**Synthetic text generation in FL.** Prior work uses public LLMs to generate synthetic texts that pretrain or warm-start smaller on-device models (Wang et al., 2023; Wu et al., 2024). PrE-Text is proposed in Hou et al. (2024), training small models on PrE-Text data generated by pretrained LLMs. Follow-up work frames private on-device learning as preference optimization to improve DP synthetic data quality (Hou et al., 2025). However, these methods are primarily designed for cross-device FL with many clients holding very small local datasets and generally rely on prompting pretrained LLMs rather than adapting them to domain shift; consequently, distribution drift and domain adaptation are not directly addressed, and limited per-client data constrain the effectiveness of any local finetuning.

## 6 CONCLUSION

We tackled the problem of generating DP synthetic text in cross-silo FL, where computational and data heterogeneity pose significant challenges. Our framework combines DP federated finetuning on strong clients with a lightweight voting mechanism from weak clients, guided by control codes. This two-phase design allows partial finetuning to capture broad patterns of the global dataset, while refinement incorporates weak-client distributions to reduce bias and mitigate the adverse effects of DP noise. Experiments on benchmark datasets under both IID and non-IID settings demonstrate that our approach consistently improves downstream utility and distributional fidelity under DP and data heterogeneity. Looking ahead, combining control codes with prompt-based methods offers a promising direction for further improving quality of synthetic data, while richer profiling strategies could enhance the role of weak clients in shaping high-fidelity synthetic datasets.

## 7 ETHICS STATEMENT

This work complies with the ICLR Code of Ethics. We use only publicly available benchmark datasets (Yelp Reviews and PubMed abstracts), with no human subjects or sensitive personal data involved. Our methods focus on privacy-preserving text generation in federated learning and do not raise additional ethical concerns beyond standard considerations for text generation models.

## 8 REPRODUCIBILITY STATEMENT

The full implementation of our framework, including DP federated finetuning and the refinement step, is provided in the supplementary materials as anonymized code. Hyperparameter choices, training configurations are included in the appendix. Dataset partitioning strategies for both IID and non-IID settings are mentioned in the main paper.

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

## A DISCUSSION ABOUT RELATED WORK

In this section, we further clarify the relationship between our paper and prior work about applying LoRA and Private Evolution (PE) related.

Our paper is orthogonal to prior work about apply LoRA in FL. LoRA-based federated adaptation is indeed highly relevant as a communication-efficient approach to LLM finetuning. However, these methods still assume that participating clients can perform local backpropagation in a timely manner; they primarily reduce the number of trainable and communicated parameters, not the *local compute requirement*. Our work explicitly targets a setting where only a subset of clients have sufficient compute for any form of local finetuning, and the remaining clients can only afford inference and lightweight similarity computations. In this sense, LoRA and related PEFT approaches are complementary to our framework: they can be used *within Phase 1* as an alternative finetuning mechanism for strong clients, while Phase 2 (DP voting from weak clients) remains necessary to incorporate low-resource participants. We include LoRA-based finetuning results for strong clients and observe that our refinement mechanism continues to provide consistent gains on top of PEFT-style updates as shown in Table A.13.

PE and prompt-based are important related work, but they target a different setting and objective than ours. The details are as follows.

Finetuning vs. no finetuning. Our work explicitly addresses scenarios with domain shift / data drift, where the generator must adapt to new domains before synthetic data are useful. We therefore focus on DP finetuning of LLMs in FL and subsequent refinement. In contrast, PE-style methods typically assume generation directly from a fixed pretrained model without federated finetuning.

Cross-silo vs. cross-device FL and privacy accounting. Our setting is cross-silo FL, where each client is an institution holding thousands–tens of thousands of samples, and we target sample-level DP. PE is developed for cross-device FL, where clients are individual users with very few samples and the target is client-level DP. The communication patterns, participation assumptions, and privacy accounting are therefore quite different.

Prompting a finetuned model vs. a purely pretrained model. In PE and its variants, a pretrained model is prompted to generate synthetic data, and substantial prompt engineering may still be required to locate the desired distribution. In our framework, we prompt a finetuned model that has already been adapted to the target domain, and control codes are used to select the appropriate conditional distribution without additional prompt tuning.

## B ADDITIONAL ALGORITHMS

Additional algorithms used in this paper are provided in Algorithms A.3, A.2, and A.4. For the analytic Gaussian mechanism, we follow Balle & Wang (2018, Algorithm 1).

---

**Algorithm A.2:** Local Voting

**Input:** Synthetic set $\tilde{D}$, local set $D_i$, control codes $C$, encoder $g(\cdot)$, neighbors $K$, DP params $(\varepsilon, \delta)$.

**Output:** DP votes $\tilde{v}_i$ of client $i$.

**1. Initialization:** $v_i \leftarrow [0, 0, \ldots, 0] \in \mathbb{R}^{|\tilde{D}|}$

**2. Embedding:** Encode all samples: $\mathbf{Z}_{\text{syn}} \leftarrow g(\tilde{D})$, $\mathbf{Z}_{\text{loc}} \leftarrow g(D_i)$.

**3. KNN voting:** For each control code $c^j$:
- Identify local embeddings $\mathbf{Z}_{\text{loc}}^j$ and synthetic embeddings $\mathbf{Z}_{\text{syn}}^j$ within $c^j$.
- For each $z \in \mathbf{Z}_{\text{loc}}^j$:
    - Find the $K$ nearest neighbors in $\mathbf{Z}_{\text{syn}}^j$ and get their indices $\mathcal{I}$.
    - Add votes to neighbors: $v_i[\mathcal{I}] \leftarrow v_i[\mathcal{I}] + 1$.

**4. Add noise:** $\tilde{v}_i \leftarrow$ `Analytical_Gaussian_Mechanism`$(v_i, \varepsilon, \delta)$.

**return** $\tilde{v}_i$.

---

---

**Algorithm A.3:** Federated Finetuning

---

**Input:** Pretrained model $\theta^0$, client set $\mathcal{C}_s$, FL rounds $R$, local iteration $\tau$, DP params
$(\varepsilon_{\text{train}}, \delta_{\text{train}})$.

**Output:** Finetuned model $\theta^\star$

**for** $r = 1$ **to** $R$ **do**

    **Server**: Broadcasts model $\theta^{r-1}$ and hyperparameters to clients in $\mathcal{C}_s$.

    **Clients in** $\mathcal{C}_s$:

        1. **Local initialization:** $\theta_i^{r-1} \leftarrow \theta^{r-1}$.

        2. **For** local iteration $e = 1, \ldots, \tau$ **do**

            (a) Iterate over minibatches $B$ of samples $(x, c^j)$ from local data $D_i$

            (b) Compute sample-level gradients $g_i = \nabla_\theta \mathcal{L}$ from loss $\mathcal{L} = \log p_{\theta_i^{r-1,e-1}}(x|c^j)$

            (c) Clip the gradients: $\tilde{g}_i \leftarrow \frac{g_i}{\max\left(1, \frac{\|g_i\|_2}{c_g}\right)}$ with gradient clipping norm $c_g$

            (d) Apply Gaussian noise to the average: $\bar{g} \leftarrow \frac{1}{B}(\sum_{i=1}^B \tilde{g}_i + \mathcal{N}(0, \sigma_s^2 c_g^2 I))$

            (e) Take a gradient step: $\theta_i^{r-1,e} \leftarrow \theta_i^{r-1,e-1} - \gamma \bar{g}$

        3. Compute local update: $\Delta_i \leftarrow \theta_i^{r-1,E} - \theta^{r-1}$.

        4. Send local updates $\Delta_k$ to Server.

    **Server**: Aggregate local updates $\Delta = \frac{1}{N} \sum_i \Delta_i$

    **Server**: Perform global update: $\theta^r \leftarrow \theta^r + \eta \Delta$.

**end**

---

**Algorithm A.4:** Sampling without Replacement

---

**Input:** Index set $\mathcal{I}^k$ denoting elements of initial synthetic dataset $\tilde{D}^k$ associated with control
codes $c^k$, sampling rate $r$, probability distribution $p^k$

**Output:** Index set $\tilde{\mathcal{I}}^k$ denoting synthetic data elements associated with control codes $c^k$
selected for inclusion in final synthetic dataset

Set count $M \leftarrow \max\{1, \lfloor r |\mathcal{I}^k| \rfloor\}$; then $M \leftarrow \min\{M, |\mathcal{I}^k|\}$.

Initialize $Q \leftarrow \mathcal{I}^k$; $\tilde{\mathcal{I}}^k \leftarrow \{\}$; $W \leftarrow \sum_{q \in Q} p_q^k$

**for** $m = 1$ **to** $M$ **do**

    Sample $i^\star$ from $R$ with $\Pr(i^\star = i) = \begin{cases} \frac{w_i}{W}, & \text{if } W > 0, \\ \frac{1}{|R|}, & \text{if } W = 0, \end{cases} \quad i \in R$

    Add $i^\star$ to $\tilde{\mathcal{I}}^k$; set $W \leftarrow W - w_{i^\star}$; remove $i^\star$ from $R$.

**end**

---

## C ADDITIONAL EXPERIMENTAL DETAILS

**Federated Finetuning.** For Yelp, we choose the max length as 128 for both finetuning and generation. This follows the setting in Yue et al. (2022) For PubMed, we choose the max length as 512 for both finetuning and generation, given that the abstracts are longer. For federated finetuning without DP, the learning rates are chosen as $\gamma = 5\text{e-}6$, $\eta = 5\text{e-}5$ for GPT-2, and $\gamma = 3\text{e-}6$, $\eta = 2\text{e-}5$ for GPT-2-large. In this case, we choose $R = 100, \tau = 50$, and batch size as 32 for GPT-2 and $R = 100, \tau = 20$ and batch size as 32 for GPT-2-large, then output the model with minimal evaluation loss. For experiments with DP, the learning rates are chosen as $\gamma = 5\text{e-}4$, $\eta = 1\text{e-}3$ for GPT-2, and $\gamma = 4\text{e-}5$, $\eta = 8\text{e-}4$ for GPT-2-large. In this case, we choose $R = 200, \tau = 50$, and batch size as 256 for GPT-2 and $R = 50, \tau = 20$, and batch size as 256 for GPT-2-large.

For federated finetuning Llama 7B with DP, the learning rates are chosen as $\eta = 0.005$ and $\gamma = 0.001$. The batch size is 64 and the clipping constant is 1. The number of local updates is 50 and the number of rounds is 50.

**Generation.** We generate $10000$ synthetic examples for Yelp and $1000$ synthetic examples for PubMed in the main results. The sampling rate in refinement is $0.2$. The temperature is $1.0$.

**Evaluation.** For classification using RoBERTa, the learning rate is 2e-5. For classification using BERT, the learning rate is 2e-5. For NER task using BERT, the learning rate 1e-4. For all experiments in evaluation, the batch size is $64$.

**Dataset.** For Yelp, we use $1.5$M examples as the global training set, $5000$ examples as test set, and another $5000$ examples as evaluation test. We use PubMed dataset prepared by Ahmad (2023). In particular, we use $45000$ examples as the global training set, $2500$ examples as test set, and another $2500$ examples as evaluation set.

**Environment.** All experiments were conducted on an NVIDIA DGX system equipped with 8 H100 GPUs (80GB memory each), 2 AMD EPYC 9654 CPUs, and 2TB of system memory. We implemented our framework in PyTorch with Hugging Face Transformers, and used the Opacus library for differentially private training. Unless otherwise specified, experiments were run in mixed-precision mode to improve efficiency, and distributed training was managed with PyTorch's native data-parallel utilities.

**DP Setting.** To ensure fairness across all clients, we assign each client the same total privacy budget $\varepsilon$, with the allocation depending on whether the client is strong or weak. We concatenate the privacy budget of each step according the basic composition theorem Dwork et al. (2014). Each client will participate in two phases. For each phase, we choose $\delta_{\text{prof}} = \delta_{\text{train}} = \delta_{\text{vote}} = \frac{1}{2N \log N}$ then we have $\delta = \frac{1}{N \log N}$.

**Strong clients ($\mathcal{C}_s$).** Strong clients participate in DP-SGD training and also provide DP control-code profiles. Their total privacy budget is

$$\varepsilon = \varepsilon_{\text{train}} + \varepsilon_{\text{prof}}.$$

To allocate more privacy to the component that most strongly affects utility (DP-SGD training), we use:
$$\varepsilon = 8: \quad \varepsilon_{\text{train}} = 6, \quad \varepsilon_{\text{prof}} = 2,$$
$$\varepsilon = 4: \quad \varepsilon_{\text{train}} = 3, \quad \varepsilon_{\text{prof}} = 1.$$

**Weak clients ($\mathcal{C}_r$).** Weak clients do not run DP-SGD; instead, they contribute through DP control-code profiling and DP voting during refinement. Thus their total privacy budget is

$$\varepsilon = \varepsilon_{\text{vote}} + \varepsilon_{\text{prof}}.$$

To maintain consistency across all clients, we use the same profiling budget $\varepsilon_{\text{prof}} = 2$ as above. The remaining privacy budget is assigned to voting:
$$\varepsilon = 8: \quad \varepsilon_{\text{vote}} = 6, \quad \varepsilon_{\text{prof}} = 2,$$
$$\varepsilon = 4: \quad \varepsilon_{\text{vote}} = 3, \quad \varepsilon_{\text{prof}} = 1,$$

These choices ensure (i) fairness (all clients receive the same total $\varepsilon$), (ii) consistency (profiling uses the same budget across strong and weak clients), and (iii) utility (the larger privacy share is allocated to training or voting, which are the components that most influence downstream performance).

## D    SYNTHETIC TEXTS AND PROMPTS

We sampled synthetic texts generated in different cases shown in listings 1–4. For the synthetic texts generated by pretrained models, we use prompts as follows.

Yelp (X and Y are substituted by the specific business category and the rating):

```
Please generate a review about business category [X] with
rating star [Y]:\n\n
```

PubMed (X is substituted by the MeSH terms covered and Y is substituted by the MeSH terms unrelated):

```
Please generate a scientific abstract on a biomedical study that
covers topics [X] but does not cover topics [Y]:\n\n
```

For the finetuned models, since we use controllable text generation, the labels of Yelp dataset and the MeSH terms of PubMed dataset are directly used as prompts.

Listing 1: Yelp synthetic examples generated by DP finetuned GPT-2 by IID $10\%$ pariciption.

```
"I've been here a lot. This place is underperforming and there isn't much
    to go back. A nurse told me to get off there, she called again to
   make sure that everything was okay. She said she could take a rest. I
    have to ask if the doctor was ever out of town. That doesn't matter.
    They were waiting on the line. My husband and I took our baby to the
    hospital to check him in there. The whole staff came out to be with
   us the whole time. Not a fan.",Business Category: Health & Medical,
   Review Stars: 2.0
```

Listing 2: Yelp synthetic examples generated by DP finetuned GPT-2 with refinement by non-IID 1 & 5 stars.

```
"This place is wonderful. Everything I ate was delicious. The food was
    great. They serve fresh and they put you in a cute box with a little
    table on it. It's perfect for the night. We ordered the veggie
    chicken salad, bacon, bacon- a must! The bacon had a little bit of
    cheese on it, but it was a nice change from everything I had. I love
    this place, it's just sooo convenient. I recommend ordering more and
    not going here.",Business Category: Bars,Review Stars: 4.0
```

Listing 3: PubMed synthetic examples generated by DP finetuned GPT-2-large by IID 10% $\mathcal{C}_s$ clients.

```
"The purpose of this study was to evaluate the effects of two different
    forms of pomegranate juice (PSJ) on glucose, insulin, and lipid
    profile in healthy human volunteers. The study participants included
    a non-insulin-dependent diabetic (NIDDM) patient, a patient with mild
     to moderate type 2 diabetes mellitus (M2DM), and a healthy, nonobese
    , nonsmoking male volunteer. Methods: The subjects consumed 10 g/d of
     PSJ for three weeks, for which they were told to drink 1 L/day of a
    glucose syrup containing 5 g of glucose (sucrose) and 100 g sucrose,
    with 1-way repeated-measures ANOVA with dietary (glucose, fructose)
    as the between-subject factor. A post hoc two-sample t-test was used
    to assess the differences between the two forms. Blood samples were
    collected before and after the dietary intervention. Plasma glucose
    and insulin and lipoprotein (C) were measured by enzymatic methods.
    Results: Results were expressed as means $\pm$ SEM, while means and
    SDs were calculated from multiple linear regression models.
    Differences were considered statistically significant at $p < 0.05$."
     Anatomy: 1, Diseases: 0, Chemicals and Drugs: 1, Named Groups: 1,
    Health Care: 0
```

Listing 4: PubMed synthetic examples generated by DP finetuned GPT-2-large with refinement by non-IID including M, N .

```
"The purpose of this study was to compare the use of a two-component,
    structured cognitive task (i.e., the Cognitive Behavioral Therapy/
    Prolonged Exposure (CBT-PE/PE) and Brief Cognitive Therapy/Prolonged
    Exposure (BCT/PE/PE) therapy, with a 30-day wait-list control group)
    in the treatment of depression with and without comorbid personality
    disorders in a sample of patients referred for evaluation of clinical
     needs for CBT and BCT. Patients were enrolled from a hospital-based
    psychiatry hospital. Thirty-five patients completed the study. CBT
    was more effective (p = 0.02) in the treatment of depression with or
    without personality disorders in comparison to BCT/PE/PE. There was
    no evidence of difference between CBT and BCT in comparison to BCT/PE
```

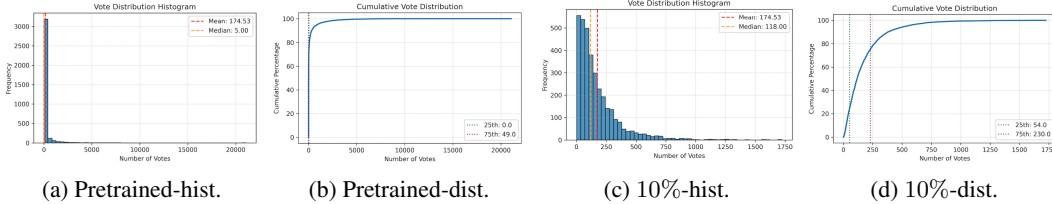

(a) Pretrained-hist.     (b) Pretrained-dist.     (c) 10%-hist.     (d) 10%-dist.

Figure A.3: The vote distributions of Yelp synthetic data generated for restaurant and rating stars 3. The left two figures are results only using the pretrained model. The right two figures are results using finetuned model with $10\%$ $\mathcal{C}_s$ clients. Before finetuning, most samples receive similar numbers of votes, limiting the effect of refinement. After DP finetuning, however, a long-tail distribution emerges: certain samples receive significantly more votes than others (Figure A.3d), reflecting the model's improved ability to generate text that aligns with the original data distribution. This concentration of votes on high-quality samples is precisely what our refinement step exploits to enhance synthetic data quality.

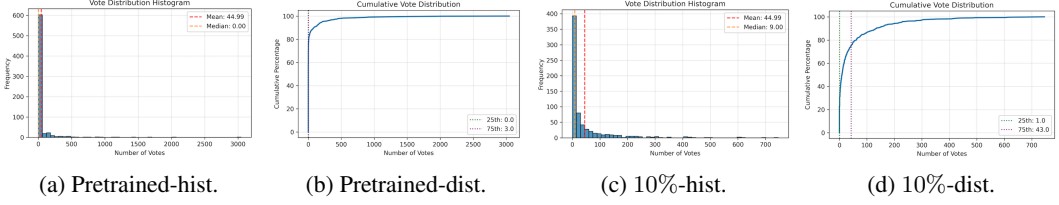

(a) Pretrained-hist.     (b) Pretrained-dist.     (c) 10%-hist.     (d) 10%-dist.

Figure A.4: The vote distributions of PubMed synthetic data generated for A, D. The left two figures are results only using the pretrained model. The right two figures are results using finetuned model with $10\%$ $\mathcal{C}_s$ clients. Similar observations as those from Figure A.3 can be obtained.

```
/PE. These results suggest that, in comparison to CBT and BCT/PE/PE,
CBT might be more efficacious than BCT/PE in the treatment of
depression and the presence of personality disorders.",Anatomy:1,
Diseases:1,Chemicals and Drugs:0,Named Groups:0,Health Care:1
```

## E  ADDITIONAL EXPERIMENTAL RESULTS

First, we examine how refinement operates in practice. Figures A.3 and A.4 show the voting distributions for synthetic data generated before finetuning and after finetuning.

Second, then we conducted ablation studies on the refinement step using Yelp dataset. For efficiency, the number of final synthetic texts is set as 5000. The results can be found in Tables A.7 and A.8

Third, additional experimental results for PubMed can be found in Tables A.10 and A.11.

## F  LLM USE

We made limited use of LLMs in preparing this manuscript. Specifically, the models were only applied at the sentence level for minor polishing to improve readability and clarity of expression. They were not involved in formulating research ideas, designing methodology, analyzing data, or generating substantive content. All conceptual, technical, and scientific contributions were made entirely by the authors.

Table A.7: Downstream results with DP Yelp synthetic texts with refinement for different sampling rates $r$ while the number of final synthetic texts is fixed. It can be seen that lower sampling rate, equivalent to higher number of initial synthetic texts, can bring better accuracy and F1 score.

|  | Acc.-1 | F1-1 | Acc.-2 | F1-2 |
|---|---|---|---|---|
| $r = 0.5$ | 0.6878 | 0.6807 | 0.5627 | 0.5389 |
| $r = 0.25$ | 0.7046 | 0.6948 | 0.5713 | 0.5561 |
| $r = 0.17$ | 0.7127 | 0.7020 | 0.5904 | 0.5678 |

Table A.8: Downstream results with DP Yelp synthetic texts with refinement for different sampling rates $K$ while other hyperparameters are fixed. Different from $r$, there is no consistent pattern as $K$ increases. One possible reason could be when the number of votes is sufficiently large, increasing votes might not be able to improve the results further.

|  | Acc.-1 | F1-1 | Acc.-2 | F1-2 |
|---|---|---|---|---|
| $K = 1$ | 0.7099 | 0.6989 | 0.5721 | 0.5496 |
| $K = 3$ | 0.6940 | 0.6891 | 0.5786 | 0.5600 |
| $K = 5$ | 0.7046 | 0.6948 | 0.5713 | 0.5561 |

Table A.9: Downstream results with DP Yelp synthetic texts with refinement for different temperature while other hyperparameters are fixed.

| Temp. | Acc.-1 | F1-1 | Acc.-2 | F1-2 |
|---|---|---|---|---|
| 0.8 | 0.7096 | 0.6968 | 0.5609 | 0.5486 |
| 1.0 | 0.7046 | 0.6948 | 0.5713 | 0.5561 |
| 1.2 | 0.6958 | 0.6891 | 0.5611 | 0.5518 |

Table A.10: Experimental results for downstream tasks using PubMed synthetic data with IID setting. Classification results for Anatomy (A), Disease (C) and Persons (M) are provided.

| Part. | $\varepsilon = \infty$ | | | | | |
|---|---|---|---|---|---|---|
|  | Acc.(A) | F1(A) | Acc.(C) | F1(C) | Acc.(M) | F1(M) |
| Pre. | 0.5280 | 0.3738 | 0.5604 | 0.4978 | 0.5708 | 0.5226 |
| 5% | 0.6000 | 0.5965 | 0.6384 | 0.6375 | 0.7204 | 0.7173 |
| 10% | 0.5960 | 0.5742 | 0.6760 | 0.6704 | 0.7796 | 0.7804 |
| 20% | 0.6468 | 0.6468 | 0.7528 | 0.7513 | 0.7900 | 0.7910 |
| 30% | 0.7180 | 0.7166 | 0.7688 | 0.7691 | 0.8056 | 0.8060 |
| 40% | 0.7204 | 0.7206 | 0.7648 | 0.7651 | 0.8152 | 0.8159 |
| Part. | $\varepsilon = 8$ | | | | | |
|  | Acc.(A) | F1(A) | Acc.(C) | F1(C) | Acc.(M) | F1(M) |
| Pre. | — | — | — | — | — | — |
| 5% | 0.5224 | 0.5099 | 0.5496 | 0.4283 | 0.5744 | 0.5293 |
| 10% | 0.5621 | 0.5155 | 0.5464 | 0.4618 | 0.5824 | 0.5471 |
| 20% | 0.5628 | 0.5296 | 0.5448 | 0.4884 | 0.6044 | 0.5489 |
| 30% | 0.5636 | 0.5321 | 0.5632 | 0.5358 | 0.6092 | 0.5904 |
| 40% | 0.5664 | 0.5392 | 0.5652 | 0.5484 | 0.6144 | 0.6064 |
| Part. | $\varepsilon = 8$ **with refinement** | | | | | |
|  | Acc.(A) | F1(A) | Acc.(C) | F1(C) | Acc.(M) | F1(M) |
| Pre. | 0.5536 | 0.4474 | 0.6736 | 0.6656 | 0.5804 | 0.5580 |
| 5% | 0.5836 | 0.5495 | 0.6828 | 0.6821 | 0.7884 | 0.7893 |
| 10% | 0.6024 | 0.5797 | 0.6896 | 0.6874 | 0.8000 | 0.8008 |
| 20% | 0.6200 | 0.6026 | 0.6984 | 0.6960 | 0.8080 | 0.8073 |
| 30% | 0.6224 | 0.6154 | 0.7180 | 0.7100 | 0.8156 | 0.8166 |
| 40% | 0.6272 | 0.6320 | 0.7220 | 0.7219 | 0.8136 | 0.8146 |

Table A.11: Experimental results for downstream tasks using PubMed synthetic data in non-IID setting. Classification results for Persons (M), and Healthcare (N) are provided in three settings where only the subset of classes list in the Bias column are represented on clients participating in federated finetuning.

| Bias | $\varepsilon = \infty$ | | $\varepsilon = \infty$ with refinement | | $\varepsilon = 8$ | | $\varepsilon = 8$ with refinement | |
|---|---|---|---|---|---|---|---|---|
| | Acc.(M) | F1(M) | Acc.(M) | F1(M) | Acc.(M) | F1(M) | Acc.(M) | F1(M) |
| **M, N** | 0.7336 | 0.7307 | 0.7408 | 0.7315 | 0.6927 | 0.6831 | 0.7134 | 0.7046 |
| **C, M, N** | 0.7316 | 0.7327 | 0.8124 | 0.8125 | 0.6932 | 0.6918 | 0.7521 | 0.7432 |
| **D, C, M, N** | 0.7744 | 0.7742 | 0.8152 | 0.8155 | 0.7326 | 0.7245 | 0.7731 | 0.7638 |

| Bias | $\varepsilon = \infty$ | | $\varepsilon = \infty$ with refinement | | $\varepsilon = 8$ | | $\varepsilon = 8$ with refinement | |
|---|---|---|---|---|---|---|---|---|
| | Acc.(N) | F1(N) | Acc.(N) | F1(N) | Acc.(N) | F1(N) | Acc.(N) | F1(N) |
| **M, N** | 0.6652 | 0.6605 | 0.7368 | 0.7256 | 0.6029 | 0.5921 | 0.6437 | 0.6325 |
| **C, M, N** | 0.7084 | 0.7089 | 0.7336 | 0.7307 | 0.6627 | 0.6523 | 0.7038 | 0.6936 |
| **D, C, M, N** | 0.7136 | 0.7141 | 0.7560 | 0.7526 | 0.6731 | 0.6642 | 0.7242 | 0.7135 |

Table A.12: Experimental results for downstream tasks using Yelp synthetic data with non-IID setting for $\varepsilon = 4$. $\mathcal{C}_s$ clients takes up 10%.

| Rating classes | Category classes | $\varepsilon = 4$ | | $\varepsilon = 4$ with refinement | |
|---|---|---|---|---|---|
| | | Acc.-2 | F1-2 | Acc.-2 | F1-2 |
| All | All | 0.6008 | 0.5460 | 0.6147 | 0.6160 |
| **1 & 3 stars** | All | 0.5124 | 0.3758 | 0.5869 | 0.5153 |
| **1 & 5 stars** | All | 0.4725 | 0.3324 | 0.5558 | 0.5558 |
| **3 & 5 stars** | All | 0.4460 | 0.2963 | 0.5014 | 0.4784 |
| **1, 3, & 5 stars** | All | 0.6003 | 0.5487 | 0.6047 | 0.6117 |

Table A.13: Experimental results for downstream tasks using Yelp synthetic data with non-IID setting with LLaMA. $\mathcal{C}_s$ clients takes up 10%.

| Rating classes | Category classes | $\varepsilon = 8$ | | $\varepsilon = 8$ with refinement | |
|---|---|---|---|---|---|
| | | Acc.-2 | F1-2 | Acc.-2 | F1-2 |
| All | All | 0.6682 | 0.6489 | 0.6772 | 0.6509 |
| **1 & 3 stars** | All | 0.6596 | 0.6220 | 0.6631 | 0.6483 |
| **1 & 5 stars** | All | 0.6346 | 0.5692 | 0.6410 | 0.6188 |
| **3 & 5 stars** | All | 0.6520 | 0.6248 | 0.6619 | 0.6310 |
| **1, 3, & 5 stars** | All | 0.6595 | 0.6399 | 0.6613 | 0.6444 |

Table A.14: Experimental results for downstream tasks using Yelp synthetic data with non-IID setting for $\varepsilon = 8$ with uniform sampling. The number of generated samples is 50000. Then 10000 samples are uniformly sampled to do downstream tasks. Results without sampling and results with refinement are included for the ease of comparison. $\mathcal{C}_s$ clients takes up 10%.

| Rating classes | Category classes | $\varepsilon = 8$ | | $\varepsilon = 8$ with uniform sampling | | $\varepsilon = 8$ with refinement | |
|---|---|---|---|---|---|---|---|
| | | Acc.-2 | F1-2 | Acc.-2 | F1-2 | Acc.-2 | F1-2 |
| All | All | 0.6280 | 0.5819 | 0.6076 | 0.5566 | 0.6326 | 0.6002 |
| **1 & 3 stars** | All | 0.5266 | 0.3895 | 0.5158 | 0.3800 | 0.6326 | 0.6002 |
| **1 & 5 stars** | All | 0.5658 | 0.4853 | 0.5623 | 0.4662 | 0.6008 | 0.5876 |
| **3 & 5 stars** | All | 0.6280 | 0.5819 | 0.6076 | 0.5566 | 0.6326 | 0.6002 |
| **1, 3, & 5 stars** | All | 0.6084 | 0.5740 | 0.5996 | 0.5544 | 0.6232 | 0.6078 |

