# OpenReview forum: "Flexible Participation for Differentially Private Synthetic Text Generation in Cross-Silo Federated Learning"
_ICLR.cc/2026/Conference — Submitted to ICLR 2026_

### Official Review · Reviewer_CFrw · 2025-10-25

**Soundness:** 3
**Presentation:** 3
**Contribution:** 2
**Rating:** 4
**Confidence:** 3

**Summary:**

The paper addresses generation of differentially private (DP) synthetic text in a cross-silo federated learning setting where clients have heterogeneous compute and data. The proposed two-stage framework lets strong (well-resourced) clients perform DP-SGD federated finetuning of a conditional generator (using control codes), while weak clients contribute via a lightweight DP voting / profiling mechanism that refines generated candidates without backpropagation. Control codes encode semantic partitions (labels/topics) and guide both generation proportions and localized voting to ensure semantically coherent refinement. Experiments on Yelp and PubMed (IID and non-IID partitions) show that (i) partial finetuning by a small fraction of clients improves over zero-shot generation, (ii) the DP voting refinement recovers much of the utility lost to DP noise and client heterogeneity, and (iii) gains hold across several downstream tasks and MAUVE / NER metrics.

**Strengths:**

- Clear problem motivation & practical relevance. Tackles a realistic cross-silo scenario: many organizations holding sizable private text, with widely varying compute budgets — a problem of immediate practical interest.
- Simple, interpretable two-phase design. Splitting work between DP federated finetuning and DP voting is elegant: it lets expensive updates be restricted to capable nodes while still incorporating the remaining clients’ distributions. The use of control codes to structure generation is sensible and easy to implement.
- Concrete algorithmic description and reproducibility effort. The paper includes pseudocode (Algorithm 1, A.2, A.3) and details on datasets, models, and hyperparameters (GPT-2, GPT-2-large, embedding model, MAUVE / downstream classifiers). Authors claim anonymized code in the supplement. These details aid reproducibility.
- Empirical evidence across IID and non-IID settings. The experiments test several participation rates (1–40% Cs), different privacy settings (ε=∞ vs ε=8), and show consistent improvements from the refinement stage across tasks and metrics (classification, MAUVE, NER F1). Tables and plots are informative.

**Weaknesses:**

- Baselines and ablations are limited. The baselines are (i) zero-shot pretrained generation and (ii) non-DP finetuning. Missing but important comparisons include: (a) PEFT / LoRA / adapter-style federated finetuning (compute-efficient finetuning that still requires backprop but is deployable), (b) other synthetic data generation / refinement approaches (e.g., preference-optimized or prompt-based DP synthesis), and (c) stronger ablations: effect of K (votes per example), sampling rate r in resampling, the sentence embedder choice, and synthetic dataset size s. These would better isolate where gains come from.
- Robustness & adversarial behavior not studied. The voting stage aggregates noisy votes from Cr clients. How robust is the procedure to malicious or biased voting (e.g., a client submitting anomalous profiles/votes)? Is there an attack model (and mitigation) — e.g., outlier detection, clipping of votes, or robust aggregation? Without such analysis, a deployment risk remains.
- Reliance on control codes and their privacy assumption. The method assumes control codes are public / non-private and that partitioning by code yields semantically coherent subsets. In practice, choosing/defining control codes can be nontrivial and may leak information if control codes correspond to sensitive labels. Please discuss sensitivity to mis-specified codes and privacy implications of broadcasting DP profiles over codes.

**Questions:**

- Baselines — why not PEFT/LoRA or prompt-based refinement baselines? Can you add comparisons to parameter-efficient federated finetuning (LoRA/adapter) or to refinement approaches that do not require voting (e.g., preference optimization, prompt-tuning with public seeds)?
- End-to-end privacy guarantee. What is the final (ϵ,δ) for the published synthetic dataset after composing DP-SGD training, profile perturbation, and vote perturbation? Show composition math or use advanced composition / moments accountant.
- Scaling to larger LLMs / real cross-silo deployments. Do you expect the refinement gains to persist for much larger generators (e.g., modern LLMs) or when evaluating tasks beyond classification/NER? Any deployment notes (latency, single-round comm overhead for Cr)?

---

> ### Author Response · Authors · 2025-12-03
> **Response to Reviewer CFrw**
>
> We thank Reviewer CFrw for the suggestions and comments.
>
> **Relationship with prior work and baselines**
>
> We would like to further clarify the relationship between our paper and prior work on LoRA and PE-related methods.
>
> Our paper is orthogonal to prior work on LoRA. LoRA-based federated adaptation (e.g., FLoRA 2024) is indeed highly relevant as a communication-efficient approach to LLM finetuning. However, these methods still assume that participating clients can perform local backpropagation in a timely manner; they primarily reduce the number of trainable and communicated parameters, not the *local compute requirement*. Our work explicitly targets a setting where only a subset of clients have sufficient compute for any form of local finetuning, and the remaining clients can only afford inference and lightweight similarity computations. In this sense, LoRA and related PEFT approaches are complementary to our framework: they can be used *within Phase 1* as an alternative finetuning mechanism for strong clients, while Phase 2 (DP voting from weak clients) remains necessary to incorporate low-resource participants. In the revised experiments, we include LoRA-based finetuning results for strong clients and observe that our refinement mechanism continues to provide consistent gains on top of PEFT-style updates.
>
> We agree that Private Evolution and prompt-based methods are important related work, but they target a different setting and objective than ours, which makes the direct comparison on accuracy less meaningful:
>
> - **Finetuning vs. no finetuning.** Our work explicitly addresses scenarios with domain shift / data drift, where the generator must adapt to new domains before synthetic data are useful. We therefore focus on DP finetuning of LLMs in FL and subsequent refinement. In contrast, PE-style methods typically assume generation directly from a fixed pretrained model without federated finetuning.
>
> - **Cross-silo vs. cross-device FL and privacy granularity.** Our setting is cross-silo FL, where each client is an institution holding thousands–tens of thousands of samples, and we target sample-level DP. PE is developed for cross-device FL, where clients are individual users with very few samples and the target is client-level DP. The communication patterns, participation assumptions, and privacy accounting are therefore quite different.
>
> - **Prompting a finetuned model vs. a purely pretrained model.** In PE and its variants, a pretrained model is prompted to generate synthetic data, and substantial prompt engineering may still be required to locate the desired distribution. In our framework, we prompt a finetuned model that has already been adapted to the target domain, and control codes are used to select the appropriate conditional distribution without additional prompt tuning.
>
> Because of these differences in scope, architecture, and privacy granularity, we view our method as complementary to PE rather than a drop-in competitor.
>
> **Ablations**
>
> We have added experiments with LLaMA using LoRA. The results (reported in Table A.13 of the revised appendix) show the same qualitative trends: (i) partial DP federated finetuning improves synthetic data quality over zero-shot generation, and (ii) the refinement stage consistently recovers utility lost to DP noise and heterogeneous participation. This supports the claim that our framework is not tied to a specific backbone architecture.
>
> In the original Appendix D, we have included ablation studies with respect to \\(r\\) and \\(K\\) with a different \\(s\\).
>
> **On robustness and adversarial voting**
>
> We agree that robustness to malicious or biased clients is an important direction, but it is orthogonal to the main focus of this work, which is DP synthetic data generation under computational and data heterogeneity in cross-silo FL. In our current formulation we assume honest-but-curious clients and do not model adversarial behavior; the voting step already includes basic normalization and DP noise, which incidentally dampen the impact of any single client. A full treatment of Byzantine or adversarial clients (e.g., via robust aggregation or explicit attack models) is beyond our present scope.

---

> ### Author Response · Authors · 2025-12-03
> **Response to Reviewer CFrw-2**
>
> **On reliance on control codes and their privacy**
>
> Our method assumes the availability of high-level control codes (e.g., labels, topics, MeSH terms) that are already present or derivable from existing metadata, similar to prior work on controllable generation (keskar2019ctrl). We view these codes as *coarse, non-sensitive attributes* that structure the data into semantically coherent subsets; the sensitive information lies in the underlying individual texts, not in the existence of, say, a “restaurant” or “cardiology” label. In our framework, what is protected by DP are *profiles and votes over control codes* (counts, similarities), which are clipped and perturbed before aggregation. Thus, even if the codes themselves are not privatized, any statistics derived from them and used by the server satisfy the same sample-level DP guarantees as the rest of the pipeline.
>
> We also do not require control codes to be perfect or highly granular: if codes are coarse or partially mis-specified, the method remains valid, and the effect is primarily a degradation toward less targeted (more global) refinement rather than a privacy violation. In practice, control codes could also be instantiated via learned clusters or topic models, which we see as a promising direction for future work to reduce manual dependence on predefined labels.
>
> **DP Composition**
>
> Please refer to our response “A more detailed DP description” to Reviewer JUqm for the detailed privacy composition.
>
> **Added Experimental Results**
>
> Please refer to our response to Reviewer JUqm for the new experiments on DP and LLaMA.

---

### Official Review · Reviewer_of8F · 2025-10-27

**Soundness:** 3
**Presentation:** 3
**Contribution:** 3
**Rating:** 4
**Confidence:** 3

**Summary:**

This paper focus on private synthetic data in FL. This research problem is meaningful when there are several downstream tasks in a federated learning setting. Authors propose a flexible participation framework that adapts to client capacities. Strong clients perform DP federated finetuning, while weak clients contribute through a lightweight DP voting mechanism that refines synthetic text. They apply control codes (e.g., labels, topics, metadata) that represent each client’s data proportions and constrain voting to semantically coherent subsets. Experiments show that the framework improves distribution alignment and downstream robustness under DP and heterogeneity.
I like the motivation of this paper, and I think this technology can be used more widely. It's just that the current experimental results and adaptation framework (synthetic data and federated learning) make me feel that they could be better.

**Strengths:**

This paper proposes a flexible participation framework for differentially private (DP) synthetic text generation in cross-silo federated learning (FL), addressing a key limitation of prior work — computational heterogeneity.

The method elegantly combines DP federated finetuning on strong clients with DP voting-based refinement from weak clients, ensuring that all participants can contribute without heavy computation.

Experiments on Yelp and PubMed datasets under both IID and non-IID settings demonstrate strong results: the approach significantly improves synthetic data quality and downstream task performance while maintaining DP guarantees. The results are systematically presented and ablated, showing consistent gains from the refinement step.

This paper is well written.

**Weaknesses:**

1.	For experiments, the privacy parameter epsilon is set as 8. Why choose this number, 8 is quite large in DP.

2.	The computational cost of the voting step and privacy accounting (ε allocation across finetuning, profiling, and voting) could be more clearly analyzed.

3.	Evaluation primarily uses GPT-2 and GPT-2-large; extending to modern instruction-tuned or open-weight LLMs could strengthen claims of scalability and generality.

4.	Figure 1 is not easy to understand and needs to be explained in the legend.

5.	The entire method relies heavily on predefined control codes (such as tags, topics, and metadata).

**Questions:**

1.	How sensitive is performance to the choice of DP budgets among the three components ((ε_train, ε_prof, ε_vote))?
2.	How much communication cost is saved compared to full FL training when the number of weak clients is large?
3.	Are there plans to evaluate with larger models (e.g., Llama 3 or Gemma) to test scalability?
4.	In Table 1 and 2, epsilon=8 is following by a downward arrow, and epsilon=8 with refinement is following by a upward arrow, why?
5.	The ultimate goal of FL is "data remains stationary, model moves", to train a high-quality global model without concentrating on the original data. If the final output is synthetic data, then once these data are generated, we can directly use the synthetic data without federated learning. Then they will face the problem of re-centralization. Although this technically avoids the direct sharing of raw data, it seems to be a "step backward" in concept as it creates a new centralized dataset. Therefore, the application value of synthetic data in federated learning is questionable.

---

> ### Author Response · Authors · 2025-12-03
> **Response to Reviewer of8F**
>
> We thank Reviewer of8F for the suggestions and comments.
>
> **Experiments on more DP**
>
> \\(\varepsilon = 8\\) is a widely used setting in the literature (de2022unlocking, flemings2024differentially) and in practical industry choices. We have added experiments with \\(\varepsilon = 4\\) as an ablation study for the privacy–utility tradeoff, which can be found in Table A.12 of the revised appendix. It can be seen that the accuracy and F1 score drop with a lower privacy budget. After refinement, we still observe gains, which demonstrate the usefulness of the refinement step when \\(\varepsilon = 4\\).
>
> **Computational cost and privacy accounting**
>
> The voting step is intentionally designed to be lightweight compared to local training. On each weak client, the main cost is computing sentence embeddings for its local samples and evaluating similarities to the synthetic candidates sharing the same control code. This involves a single forward pass through a frozen encoder and a \\(k\\)-NN (or similarity) computation, without any backpropagation or multiple communication rounds. In particular, voting is performed *once* per refinement phase, whereas DP-SGD finetuning on strong clients requires repeated local updates over many rounds.
>
> **Results using other models**
>
> We have added experiments with LLaMA using LoRA. The results (reported in Table A.13 of the revised appendix) show the same qualitative trends: (i) partial DP federated finetuning improves synthetic data quality over zero-shot generation, and (ii) the refinement stage consistently recovers utility lost to DP noise and heterogeneous participation. This supports the claim that our framework is not tied to a specific backbone architecture.
>
> **Communication cost**
>
> When the number of weak clients is large, the saved communication cost is proportional to the number of weak clients. This is because (1) weak clients do not have to send model updates periodically, and (2) voting profiles are sent only once and the communication cost is much smaller than that of model updates.
>
> **About Table 1**
>
> In Table 1, the downward arrow indicates that after applying DP, the overall trend of accuracy and F1 scores is decreasing. The upward arrow indicates that after refinement, the overall trend of accuracy and F1 scores is increasing.
>
> **On synthetic data and “re-centralization” in FL**
>
> FL is primarily a framework for training under *privacy and locality constraints*, rather than a prohibition on any form of central artifact. The saying “data remains stationary, model moves” reflects regulatory limits on sharing *raw* data. In our setting, the released object is a DP-guaranteed synthetic dataset rather than original records; by construction, this release satisfies a formal \\((\varepsilon,\delta)\\)-DP guarantee and can be reused for multiple downstream tasks without further privacy cost. Thus, while synthetic data are centrally accessible, they are not equivalent to re-centralizing the raw siloed data, and in practice they complement FL by avoiding repeated FL runs (and repeated privacy spending) for each downstream task.
>
> **Added Experimental Results**
>
> Please refer to our response to Reviewer JUqm for the new experiments on DP and LLaMA.
>
> **References:**
>
> - De, Soham, et al. “Unlocking high-accuracy differentially private image classification through scale.” arXiv preprint arXiv:2204.13650 (2022).
> - Flemings, James, Meisam Razaviyayn, and Murali Annavaram. “Differentially private next-token prediction of large language models.” arXiv preprint arXiv:2403.15638 (2024).

---

### Official Review · Reviewer_FP3m · 2025-11-01

**Soundness:** 3
**Presentation:** 3
**Contribution:** 3
**Rating:** 6
**Confidence:** 3

**Summary:**

This paper proposes a two-phase framework for **Differentially Private (DP)** synthetic text generation in **cross-silo federated learning (FL)**, designed to handle **heterogeneous client resources**.

The key idea is to allow **flexible participation**:
- **Strong clients** (with sufficient compute) perform **DP-SGD finetuning** of a pretrained LLM to adapt it to domain-specific data.
- **Weak clients** (unable to train locally) contribute via a **lightweight DP voting mechanism** that refines the synthetic data distribution, ensuring representation of all clients.

Control codes (labels, topics, metadata) guide both finetuning and refinement, partitioning data into coherent subsets.

Experiments on **Yelp Reviews** and **PubMed abstracts** demonstrate that this approach improves **distributional alignment and downstream performance** of generated synthetic data, particularly under DP constraints and heterogeneous participation. Refinement with weak-client voting mitigates the utility loss typically induced by DP noise and biased finetuning.

**Strengths:**

- **Novel integration of heterogeneous participation and DP**: The flexible two-phase structure is a natural and elegant way to involve all clients under compute constraints.
- **Sound motivation**: Addresses a practical gap between FL for large models and realistic cross-silo deployment, where client capacity varies widely.
- **Technical clarity**: The description of Algorithm 1 and control-code-based conditioning is detailed and well-structured.
- **Empirical evaluation**:
  - Compares baselines with and without DP, and with/without refinement.
  - Uses multiple datasets and metrics (classification accuracy/F1, MAUVE, NER).
  - Results consistently show refinement improves performance under DP, especially with few strong clients.

**Weaknesses:**

1. **Privacy accounting and budgets**
   - The paper applies separate ε = 8 budgets for training and refinement, but it’s unclear whether these compose into a total DP guarantee or are treated independently.
   - The choice ε = 8 is relatively high; discussion of lower-ε performance or practical implications would strengthen credibility.
   - Clarify the full \((\varepsilon_\text{total}, \delta_\text{total})\) budget.

2. **Evaluation of privacy–utility trade-off**
   - All experiments use ε = 8; it would be valuable to show at least one lower ε (e.g., 4 or 2) to demonstrate robustness to stricter privacy.
   - Plotting performance as a function of ε would better illustrate the trade-off curve.

3. **Refinement mechanism interpretability**
   - The “DP voting” phase uses noisy aggregated similarity scores. It would help to explain how KNN-based votes interact with control codes and whether Gaussian noise biases toward majority groups.
   - Are weak clients’ votes weighted equally regardless of dataset size?

4. **Baselines and ablations**
   - The “voting” mechanism could be compared against a simpler aggregation (e.g., uniform resampling or non-DP voting) to isolate its effect.
   - Clarify whether “pretrained + voting” (without any finetuning) was tested.

5. **Conceptual framing**
   - While the “control code” abstraction is central, it’s borrowed from prior controllable generation work; the novelty lies in its federated adaptation.
   - The term *Flexible Participation* might overstate generality: the method still assumes clients can be cleanly partitioned into strong/weak and have known control-code profiles.

6. **Relation to prior work** Could better contrast with recent LoRA-based federated adaptation (e.g., FLoRA 2024) and DP synthetic text methods that do global aggregation rather than federated (e.g. private evolution (voting), DP fine tuning with LoRA, PATE based methods)

**Questions:**

NA

---

> ### Author Response · Authors · 2025-12-03
> **Response to Reviewer FP3m**
>
> We thank Reviewer FP3m for suggestions and comments.
>
> **Privacy accounting and budgets**
>
> Please refer to our response “A more detailed DP description” to Reviewer JUqm for the detailed privacy composition.
>
> **Privacy–utility tradeoff**
>
> We have added experiments with \\(\varepsilon=4\\) as an ablation study for the privacy–utility tradeoff, which can be found in Table A.12 of the revised appendix. It can be seen that the accuracy and F1 score drop with a lower privacy budget. After refinement, we still observe gains, which demonstrates the usefulness of the refinement step when \\(\varepsilon=4\\).
>
> **Refinement mechanism interpretability**
>
> The local voting process is detailed in Algorithm A.2 in Appendix A. Briefly, voting is performed *within* each control code. For a given control code \\(c^k\\), each weak client compares its local samples tagged with \\(c^k\\) to the synthetic candidates also tagged with \\(c^k\\) using a \\(k\\)-NN similarity in embedding space, and accumulates integer vote counts for those candidates. There is no cross-code interaction: control codes define independent refinement pools, so voting only refines semantically coherent subsets.
>
> Formally, for a fixed control code, let \\(v_i\\) denote the total (non-private) vote count received by synthetic sample \\(i\\), and let \\(z_i\\) denote the DP noise added to that count (we use i.i.d. Gaussian noise). The privatized scores are
> \\[
> \tilde{v}_i \\;=\\; v_i + z_i,
> \\]
> and the sampling probability for candidate \(i\) is obtained by normalizing within the same control code:
> \\[
> p_i \\;=\\; \frac{\tilde{v}_i}{\sum_j \tilde{v}_j}
> \\;=\\; \frac{v_i + z_i}{\sum_j v_j + \sum_j z_j}.
> \\]
> Because normalization is performed separately for each control code, there is no bias toward majority *across* codes.
>
> Regarding the weighting of weak clients, votes are defined at the *sample* level rather than the client level: each local example contributes equally to the vote counts. As a result, clients with more local data naturally cast more total votes, so their influence is proportional to their share of the underlying population, which is consistent with the goal of approximating the global data distribution. We will clarify these points in the revised version.
>
> **Baselines and ablations**
>
> We added experiments with uniform sampling in Table A.14 of the revised appendix. It can be seen that the accuracies and F1 scores with uniform sampling are close to or even worse than those without sampling, and much worse than with refinement, which shows the effectiveness of the refinement step.
>
> Pretrained+voting is included in the main paper. In Tables 1 and 2, the results for “Pre.” under “with refinement” correspond to Pretrained+voting.
>
> **Conceptual framing (control codes and flexible participation)**
>
> We agree that the main novelty regarding control codes lies in their *federated* adaptation rather than in the control-code abstraction itself, which is borrowed from prior controllable generation work. Our contribution is to integrate control codes into a two-phase FL pipeline in which they (i) structure DP finetuning on strong clients, and (ii) provide a common representation of heterogeneous client distributions via DP control-code profiles and code-wise refinement. This coupling is what enables weak clients to influence the generator without performing any local training.
>
> Regarding the term *flexible participation*, our intention is to emphasize that participation is adapted to clients’ computational capacities rather than restricted to those capable of full local training. The strong/weak partition is not assumed to be fixed or intrinsic; it can be chosen based on system-level constraints such as tolerated training delay or resource availability. In the experiments, we explicitly study a conservative regime where only a small fraction of clients satisfy the efficiency requirements for local DP finetuning, and the remaining clients participate through the lightweight voting mechanism.

---

> ### Author Response · Authors · 2025-12-03
> **Response to Reviewer FP3m-2**
>
> **Added Experimental Results**
>
> Please refer to our response to Reviewer JUqm for the new experiments on DP and LLaMA. The experimental results for uniform sampling are as follows:
>
> **Table 3. Experimental results for downstream tasks using Yelp synthetic data with non-IID setting for \\(\varepsilon = 8\\) with uniform sampling. The number of generated samples is 50,000. Then 10,000 samples are uniformly sampled to do downstream tasks. Results without sampling and with refinement are included for ease of comparison. \\(\mathcal{C}_s\\) clients take up \\(10\\%\\).**
>
> | **Rating classes**        | **Category classes** | **\\(\varepsilon = 8\\)** Acc.-2 | **\\(\varepsilon = 8\\)** F1-2 | **\\(\varepsilon = 8\\) with uniform sampling** Acc.-2 | **\\(\varepsilon = 8\\) with uniform sampling** F1-2 | **\\(\varepsilon = 8\\) with refinement** Acc.-2 | **\\(\varepsilon = 8\\) with refinement** F1-2 |
> |---------------------------|----------------------|---------------------------------|------------------------------|-----------------------------------------------------|----------------------------------------------------|-----------------------------------------------|----------------------------------------------|
> | All                       | All                  | 0.6280                          | 0.5819                       | 0.6076                                              | 0.5566                                             | 0.6326                                        | 0.6002                                       |
> | **1 & 3 stars**           | All                  | 0.5266                          | 0.3895                       | 0.5158                                              | 0.3800                                             | 0.6326                                        | 0.6002                                       |
> | **1 & 5 stars**           | All                  | 0.5658                          | 0.4853                       | 0.5623                                              | 0.4662                                             | 0.6008                                        | 0.5876                                       |
> | **3 & 5 stars**           | All                  | 0.6280                          | 0.5819                       | 0.6076                                              | 0.5566                                             | 0.6326                                        | 0.6002                                       |
> | **1, 3, & 5 stars**       | All                  | 0.6084                          | 0.5740                       | 05996                                               | 0.5544                                             | 0.6232                                        | 0.6078                                       |
>
> ---

---

### Official Review · Reviewer_JUqm · 2025-11-01

**Soundness:** 2
**Presentation:** 3
**Contribution:** 3
**Rating:** 4
**Confidence:** 5

**Summary:**

The paper proposes a flexible participation framework in the cross silo setting, where some clients may be imbalanced in terms of resources. In particular, some clients may have enough infrastructure and GPUs to run DPSGD on their own data, and some clients may not have enough compute power. So the method proposes to do local DPSGD on clients with compute budget, and use a method similar to Private Evolution to condition synthetic data generations towards the data on the clients which do not have much compute budget. In this way they are able to balance the benefits of training while also leveraging the data of clients that cannot train. They support their algorithm with a set of experimental evaluations on the GPT2 model.

**Strengths:**

- The algorithm does make sense. While it is not the most elegant thing, it resembles production-ready algorithms in industry which may be a combination of several different algorithms to handle different regimes (the algorithm proposed is essentially FedAvg + Private Evolution).
- The participation model helps include a wide variety of clients, which allows synthetic data to represent a wide diversity of data
- Experimental ablations are decent, with evaluations across different epsilon values, datasets, and strong client participation. They demonstrate that the refinement step is important in getting good performance

**Weaknesses:**

- Could be more explicit about the privacy composition, there are multiple steps and it is unclear how they are composed
- The algorithm is mostly a simple combination of two existing ones (FedAvg, Private Evolution), there isn't a major methodological breakthrough. However this is not something I would hold against the paper.
- The evaluation could be more comprehensive. For example, they could evaluate more models outside of GPT2. Second, they should compare against prior work better. The real baseline to compare against is not just the zero-shot pretrained model. It also includes the other methods mentioned in section 5, such as Private Evolution and its variants. The papers in the related work have their own evaluations against standard datasets, so the paper should run its method on those datasets, get the numbers, and compare against the results reported in those papers. The other baseline to compare against is pure FedAvg, which should be better than the proposed method but again this evaluation is needed for comparison.

Overall I think this is a promising paper, but feels somewhat incomplete at the moment. For it to be published at a venue like ICLR, I would want to see solid performance gains vs Private Evolution and its follow ups.

**Questions:**

See weaknesses

---

> ### Author Response · Authors · 2025-12-03
> **Response to Reviewer JUqm**
>
> We thank reviewer JUqm for the suggestions and feedbacks.
>
> 1. **A more detailed DP description.**
>
> To ensure fairness across all clients, we assign each client the same total privacy budget \\(\varepsilon\\), with the allocation depending on whether the client is strong or weak. We compose the privacy budget of each step according to the basic composition theorem in dwork2014algorithmic. Each client will participate in two phases. For each phase, we choose
> \\(\delta_{\text{prof}} = \delta_{\text{train}} = \delta_{\text{vote}} = \frac{1}{2N\log N}\\), then we have \\(\delta = \frac{1}{N\log N}\\).
>
> **Strong clients (\\(\mathcal{C}_s\\)).**
> Strong clients participate in DP-SGD training and also provide DP control-code profiles. Their total privacy budget is
> \\[
> \varepsilon = \varepsilon_{\mathrm{train}} + \varepsilon_{\mathrm{prof}}.
> \\]
> To allocate more privacy to the component that most strongly affects utility (DP-SGD training), we use:
> \\[
> \varepsilon = 8:
> \qquad \varepsilon_{\mathrm{train}} = 6,\quad \varepsilon_{\mathrm{prof}} = 2,
> \\]
> \\[
> \varepsilon = 4:
> \qquad \varepsilon_{\mathrm{train}} = 3,\quad \varepsilon_{\mathrm{prof}} = 1.
> \\]
>
> **Weak clients (\\(\mathcal{C}_r\\)).**
> Weak clients do not run DP-SGD; instead, they contribute through DP control-code profiling and DP voting during refinement. Thus their total privacy budget is
> \\[
> \varepsilon = \varepsilon_{\mathrm{vote}} + \varepsilon_{\mathrm{prof}}.
> \\]
> To maintain consistency across all clients, we use the same profiling budget \(\varepsilon_{\mathrm{prof}} = 2\) as above. The remaining privacy budget is assigned to voting:
> \\[
> \varepsilon = 8:
> \qquad \varepsilon_{\mathrm{vote}} = 6,\quad \varepsilon_{\mathrm{prof}} = 2,
> \\]
> \\[
> \varepsilon = 4:
> \qquad \varepsilon_{\mathrm{vote}} = 3,\quad \varepsilon_{\mathrm{prof}} = 1.
> \\]
>
> These choices ensure (i) fairness (all clients receive the same total \(\varepsilon\)), (ii) consistency (profiling uses the same budget across strong and weak clients), and (iii) utility (the larger privacy share is allocated to training or voting, which are the components that most influence downstream performance).
>
> ---
>
> 2. **About “The algorithm is mostly a simple combination of two existing ones…”.**
>
> While our framework indeed builds on standard components (federated optimization and DP refinement), it is not a simple composition of FedAvg and Private Evolution (PE); the main novelty lies in how these pieces are adapted and coupled for the cross-silo, heterogeneous FL setting.
>
> First, a key contribution of our work is an effective way to connect DP LLM finetuning and synthetic text generation in FL via control codes. Control codes are used consistently in both phases: (i) to structure conditional finetuning on strong clients and (ii) to represent local distributions of all clients (including weak ones) through control-code profiles, which then guide how synthetic samples are allocated and refined. Without this control-code–based coupling, it would be difficult to efficiently use the finetuned model to generate synthetic data that reflects heterogeneous client distributions.
>
> Second, our voting/refinement mechanism is designed specifically for cross-silo FL with computational heterogeneity. In our setting, weak clients participate through a single-round, DP voting step that filters and reweights samples within each control code, without any backward passes or iterative interaction. In contrast, PE-style methods typically rely on multiple rounds of interactive refinement and are not designed to handle the strong/weak client split and cross-silo participation constraints we focus on. Our framework explicitly targets the case where only a subset of clients can finetune, and the rest can only afford lightweight, one-shot participation.
>
> **More models beyond GPT-2.**
> In addition to GPT-2 / GPT-2-large, we have run new experiments with a stronger open-weight LLM (LLaMA-based) as the generator. The results (reported in Table A.13 of the revised appendix) show the same qualitative trends: (i) partial DP federated finetuning improves synthetic data quality over zero-shot generation, and (ii) the refinement stage consistently recovers utility lost to DP noise and heterogeneous participation. This supports the claim that our framework is not tied to a specific backbone architecture.

---

> ### Author Response · Authors · 2025-12-03
> **Response to Reviewer JUqm-2**
>
> 3. **On Private Evolution (PE) as a baseline.**
>
> We agree that Private Evolution and its variants are important related work, but they target a different setting and objective than ours, which makes the direct comparison on accuracy less informative:
>
> - **Finetuning vs. no finetuning.** Our work explicitly addresses scenarios with domain shift / data drift, where the generator must adapt to new domains before synthetic data are useful. We therefore focus on DP finetuning of LLMs in FL and subsequent refinement. In contrast, PE-style methods typically assume generation directly from a fixed pretrained model without federated finetuning.
>
> - **Cross-silo vs. cross-device FL and privacy granularity.** Our setting is cross-silo FL, where each client is an institution holding thousands–tens of thousands of samples, and we target sample-level DP. PE is developed for cross-device FL, where clients are individual users with very few samples and the target is client-level DP. The communication patterns, participation assumptions, and privacy accounting are therefore quite different.
>
> - **Prompting a finetuned model vs. a purely pretrained model.** In PE and its variants, a pretrained model is prompted to generate synthetic data, and substantial prompt engineering may still be required to locate the desired distribution. In our framework, we prompt a finetuned model that has already been adapted to the target domain, and control codes are used to select the appropriate conditional distribution without additional prompt tuning.
>
> Because of these differences in scope, architecture, and privacy granularity, we view our method as complementary to PE rather than a drop-in competitor. We see adapting PE-style methods to our cross-silo, sample-level DP setting as an interesting direction for future work, and complementary to the framework proposed here.
>
> ---
>
> 4. **About pure FedAvg with all clients as a baseline**
>
> Due to time and resource constraints within the rebuttal period, we were unable to run a full pure-FedAvg baseline. However, our non-DP finetuning results already approximate the behavior of such an upper bound; we view a full pure-FedAvg baseline as valuable future work..
>
> ---

---

> ### Author Response · Authors · 2025-12-03
> **Response to Reviewer JUqm-3**
>
> **Added Experimental Results**
>
> **Table 1. Experimental results for downstream tasks using Yelp synthetic data with non-IID setting for \\(\varepsilon = 4\\). \\(\mathcal{C}_s\\) clients take up \\(10\\%\\).**
>
> | **Rating classes**        | **Category classes** | **\\(\varepsilon = 4\\)** Acc.-2 | **\\(\varepsilon = 4\\)** F1-2 | **\\(\varepsilon = 4\\) with refinement** Acc.-2 | **\\(\varepsilon = 4\\) with refinement** F1-2 |
> |---------------------------|----------------------|---------------------------------|------------------------------|-----------------------------------------------|----------------------------------------------|
> | All                       | All                  | 0.6008                          | 0.5460                       | 0.6147                                        | 0.6160                                       |
> | **1 & 3 stars**           | All                  | 0.5124                          | 0.3758                       | 0.5869                                        | 0.5153                                       |
> | **1 & 5 stars**           | All                  | 0.4725                          | 0.3324                       | 0.5558                                        | 0.5558                                       |
> | **3 & 5 stars**           | All                  | 0.4460                          | 0.2963                       | 0.5014                                        | 0.4784                                       |
> | **1, 3, & 5 stars**       | All                  | 0.6003                          | 0.5487                       | 0.6047                                        | 0.6117                                       |
>
>
> **Table 2. Experimental results for downstream tasks using Yelp synthetic data with non-IID setting with LLaMA. \\(\mathcal{C}_s\\) clients take up \\(10\\%\\).**
>
> | **Rating classes**        | **Category classes** | **\\(\varepsilon = 8\\)** Acc.-2 | **\\(\varepsilon = 8\\)** F1-2 | **\\(\varepsilon = 8\\) with refinement** Acc.-2 | **\\(\varepsilon = 8\\) with refinement** F1-2 |
> |---------------------------|----------------------|---------------------------------|------------------------------|-----------------------------------------------|----------------------------------------------|
> | All                       | All                  | 0.6682                          | 0.6489                       | 0.6772                                        | 0.6509                                       |
> | **1 & 3 stars**           | All                  | 0.6596                          | 0.6220                       | 0.6631                                        | 0.6483                                       |
> | **1 & 5 stars**           | All                  | 0.6346                          | 0.5692                       | 0.6410                                        | 0.6188                                       |
> | **3 & 5 stars**           | All                  | 0.6520                          | 0.6248                       | 0.6619                                        | 0.6310                                       |
> | **1, 3, & 5 stars**       | All                  | 0.6595                          | 0.6399                       | 0.6613                                        | 0.6444                                       |
> ---

---

### Author Response · Authors · 2025-12-03
**General Responses**

We thank the reviewers and AC for their careful evaluation and feedback. All reviewers agree that the paper’s **problem setting is meaningful**, the **method is sound and clearly described**, and the **motivation (cross-silo FL with heterogeneous compute + DP synthetic text) is practically relevant**. The main concerns center around (i) **clarity of DP accounting**, (ii) **privacy–utility tradeoffs**, and (iii) **experimental coverage and baselines**, rather than the validity of the approach.

In the revised version and rebuttal, we have:

- **Clarified DP accounting and total privacy guarantees.**
  We now explicitly describe how \\(\varepsilon_{\text{train}}, \varepsilon_{\text{prof}}, \varepsilon_{\text{vote}}\\) compose per client (strong vs. weak), with a consistent \\(\delta\\), and reference the specific composition theorem used.

- **Added experiments for stricter DP budgets (\\(\varepsilon = 4\\)).**
  We report new results under \\(\varepsilon = 4\\) (Table A.12), showing that while utility decreases as expected, the **refinement step continues to provide consistent gains**, confirming that our mechanism is effective even under stronger privacy.

- **Added experiments with a stronger open-weight LLM (LLaMA + LoRA).**
  We include new experiments using a LLaMA-based generator with LoRA for DP finetuning (Table A.13). The same qualitative trends hold:
  (i) partial federated finetuning improves synthetic data quality over zero-shot generation, and
  (ii) DP refinement improves the utility under DP noise and heterogeneous participation.
  This supports the claim that our framework is **not tied to GPT-2** and is compatible with modern PEFT-style finetuning.

- **Added a uniform sampling baseline to isolate the effect of voting.**
  We compare our DP voting-based refinement with simple uniform resampling (Table A.14). Uniform sampling performs close to or worse than no sampling, and **substantially worse than our refinement**, demonstrating that the improvement is not merely due to downsampling but to the structured DP voting mechanism.

- **Clarified the relationship to Private Evolution and LoRA-based FL methods.**
  We explain that PE-style methods generally assume:
  (i) generation from a fixed pretrained model (no domain-adaptive finetuning),
  (ii) cross-device FL with client-level DP, and
  (iii) iterative interactive refinement.
  By contrast, our setting is **cross-silo, sample-level DP** with **domain shift**, and we explicitly handle a **strong/weak client split** with one-shot lightweight participation from weak clients. We also clarify that LoRA-based FL is complementary: it can replace Phase 1 finetuning on strong clients, but **cannot substitute Phase 2**, which is designed to incorporate weak clients that cannot backpropagate at all.

- **Clarified the role and assumptions on control codes.**
  We emphasize that control codes are **non-sensitive attributes** (labels, topics, MeSH terms, or derived clusters) used to structure heterogeneity, not to erase it. Their profiles and votes are protected under DP.


We hope that these additions and clarifications address the reviewers’ main concerns regarding privacy accounting, robustness across privacy budgets, experimental coverage, and the positioning of our method relative to prior work. We believe the revised version presents a clearer and more thoroughly validated picture of the framework’s strengths and limitations.

---

### Meta-Review · Area_Chair_4vuL · 2025-12-13

**Summary:**

The paper proposes a framework for differentially private federated learning in cross-silo settings where clients have heterogeneous computational resources. Reviewers were positive about the motivation, proposed framework and method, and some aspects of the empirical evaluation. Concerns focused on other key aspects of the empirical evaluation like baselines, ablations and the LLMs used, as well the relatively liberal setting of privacy budget. The rebuttals addressed these concerns for the most part by adding more experimental results. While this appreciated, the scope of results added might be considered more appropriate for a resubmission. The recommendation is therefore to revise and resubmit.

Note to the authors: The bibliography reference to FlexLora appears to have the wrong author list (to be very clear: this is a common manual error, and no AI misuse is being flagged here; this note is only for the authors' benefit).

**Reviewer Concerns:**

See above.

**Reviewer Scores:**

The initial reviews generally marked the paper as a borderline but below-the-bar paper. The rebuttal might have flipped some weak rejects to weak accepts, but in light of the scope of criticism and the scope of rebuttal additions needed to address them, it is unlikely the result would have changed to a strong consensus in favor of the paper that would have changed the outcome. It is recommended to incorporate the extended results from the rebuttal into a revised manuscript toward a resubmission.

---

### Decision · Program_Chairs · 2026-01-26

Reject